# Directing T-Cell Immune Responses for Cancer Vaccination and Immunotherapy

**DOI:** 10.3390/vaccines9121392

**Published:** 2021-11-25

**Authors:** Peter Lawrence Smith, Katarzyna Piadel, Angus George Dalgleish

**Affiliations:** Institute of Infection and Immunity, St. Georges University of London, London SW17 0RE, UK; kpiadel@sgul.ac.uk (K.P.); dalgleis@sgul.ac.uk (A.G.D.)

**Keywords:** immunotherapy, T-cell, vaccine, cancer, checkpoint inhibition, microbiome, metabolism

## Abstract

Cancer vaccination and immunotherapy revolutionised the treatment of cancer, a result of decades of research into the immune system in health and disease. However, despite recent breakthroughs in treating otherwise terminal cancer, only a minority of patients respond to cancer immunotherapy and some cancers are largely refractive to immunotherapy treatment. This is due to numerous issues intrinsic to the tumour, its microenvironment, or the immune system. CD4^+^ and CD8^+^ αβ T-cells emerged as the primary effector cells of the anti-tumour immune response but their function in cancer patients is often compromised. This review details the mechanisms by which T-cell responses are hindered in the setting of cancer and refractive to immunotherapy, and details many of the approaches under investigation to direct T-cell function and improve the efficacy of cancer vaccination and immunotherapy.

## 1. Introduction

Cancer vaccination aims to induce antigen specific T-cell based cellular immunity capable of targeting and clearing tumour cells. T-cell activation typically requires three signals; presentation of an epitope on a human leukocyte antigen (HLA) expressed by antigen presenting cells (APC), costimulation by receptors on the APC, and cytokine signalling by cytokines such as interleukin (IL)-12. The advent of immunotherapies such as immune checkpoint blockade (ICB) and T-cell receptor (TCR) or chimeric antigen receptor (CAR) T-cell adoptive therapy represent types of ‘vaccination’ that raise antigen specific T-cell responses but without therapeutically administering antigen, which is instead provided endogenously by the tumour itself. In each case the functional state of T-cell immunity is a determining factor in the success or failure of vaccination. The T-cell immune system changes throughout life, from immaturity during early years to increasing senescence later in life, whilst encountering daily antigenic challenges. These challenges range from infectious agents capable of causing acute, chronic, or latent infection and noninfectious challenges including transformed cells, self-antigen, and allergens. T-cells must accurately interrogate and interpret each of these challenges and do so often in the context of some degree of immune suppression or inflammation.

Numerous factors influence the functional state, or ‘set point’ of T-cell immunity and thus its ability to respond to cancer. These factors include age, diet, our microbiomes, prior infection and vaccination, underlying inflammation, and tumour-intrinsic immune suppression. To respond to tumorigenesis, but without generating excessive or dysfunctional immune responses, T-cells must be effectively primed before activation to generate a strong effector response and calibrated after activation to promote the transition to T-cell memory that results in efficacious Th1 based immunity. This is best achieved through the modulation of epigenetic and transcriptional programs to effect metabolism, differentiation, and survival. In this review, the factors influencing T-cell dysfunction will be described followed by the agents and strategies that are being studied to potentiate T-cell vaccination and immunotherapy. While numerous strategies involve the maturation of APCs through the use of ligands for pathogen recognition receptors, such as toll-like receptors, or deplete or block immune suppressive cells, such as myeloid-derived suppressor cells (MDSC), T-regulatory cells (T-regs) and tumour-associated macrophages or directly targeting tumour cells, the focus in this review will be on direct T-cell priming.

## 2. T-Cell Dysfunction

T-cell Immune dysfunction can manifest in numerous ways (Table 1). Chronic viral infection or cancer can induce T-cell exhaustion and apoptosis whilst T-cell deficiency can be caused by infection, immune senescence, or inflammation. In the setting of cancer, the tumour microenvironment (TME) is profoundly immunosuppressive both in the generation of a hypoxic state, metabolic acidosis and glucose competition and the evolution of mechanisms selected to recruit leukocytes and skew their development towards immune suppressive cells such as MDSC, T-regs, or TAMs, which produce cytokines such as IL-10, Vascular endothelial growth factor (VEGF), Transforming growth factor (TGF)-β, the enzyme Indoleamine 2,3-dioxygenase, arginases, or reactive oxygen species (ROS). Another method of immune suppression involves the expression of immunological checkpoints such as Programmed cell death protein ligand (PDL)-1, cytotoxic T-lymphocyte-associated protein (CTLA)-4, Galectin-9 or Fibrinogen Like-1 which are expressed on either immune suppressive cells, tumour cells, or both [1]. These checkpoints, intended to act as breaks for chronic antigenic stimulation, induce an exhausted and metabolically deficient phenotype in anti-tumour T-cells. The resulting T-cell dysfunction is responsible for the failure of anti-tumour immunity [2] and has similarities with the exhaustion associated with chronic viral infection or the T-cell dysfunction associated uncontrolled acute infections such as severe COVID-19 [3,4]. Addressing these processes is essential to treat cancer with T-cell-based vaccination or immunotherapy.

**Exhaustion.** CD8^+^ T-cell exhaustion is characterised by the expression of checkpoint receptors, including Programmed cell death protein (PD)-1, Lymphocyte activation gene (LAG)-3, and T-cell immunoglobulin and mucin-domain containing (TIM)-3, and the loss of effector functions including cytokine expression and cytotoxic effector molecules, such as granzyme and perforin [5,6] associated with altered usage of several transcription factors. The transcription factors involved in the establishment of exhaustion follow a program of activation [7]. This starts with the altered expression of the transcription factors T-box expressed in T cells (T-Bet) and Eomesodermin (EOMES), which are associated with differentiation of early effector T-cells and expression of PD-1, which, together with EOMES, subsequently acts to execute a PD-1 dependent exhaustion program. These functionally exhausted precursors are characterised by a T-cell factor (TCF)-1^+^, PD-1^+^ TIM-3^−^ phenotype. Finally, sustained TCR activation of calcineurin results in the translocation of nuclear factor of activated T-cells (NFAT) into the nucleus which can subsequently mediate T-cell exhaustion via the expression of the transcription factors Thymocyte selection associated high mobility group box (TOX) and Nuclear Receptor Subfamily 4 Group A Member (NR4A).

This state of exhaustion is thought to be initiated early during tumorigenesis [8], and it can act upon CD8^+^ T-cells, CAR T-cells or TCR transgenic T-cells. Ligation of inhibitor receptors such as PD-1 acts to inhibit TCR signalling and suppress AKT activation and mTOR activity and induce a switch from glycolysis, needed for effector T-cell function, to fatty acid oxidation (FAO) [9]. However, these exhausted cells may regain functionality upon the therapeutic use ICB, such as the anti-PDL-1 monoclonal antibody Atezolizumab and anti-PD-1 Nivolumab or Pembrolizumab which have proved particularly effective against cancers expressing mutated ‘neoantigen’ or exhibiting inflamed tumour microenvironments (TME) such as Melanoma for which anti-PD-1 Immunotherapy is now a first-line treatment [10]. Anti-PD-1 based ICB is now a first- or second-line treatment, alone or in combination for a number of cancers including non-small cell lung cancer (NSCLC) and renal cell carcinoma (RCC), however, many patients still do not respond to ICB, and some cancers are largely resistant. In such cases, T-cells exhibit a terminally exhausted phenotype, and their function cannot be restored by ICB. For example, PD-1+, CD38+ CD8^+^ T-cells in the setting of pancreatic cancer fail to respond to anti-PD-1 therapy [11]. This is due to inadequate priming prior to PD-1 blockade since improved priming with tumour antigen prior to PD-1 blockade rescues T-cell effector functions. In line with this, a similar population of CD38+ CD101+ PD1+ CD8^+^ T-cells cells are associated with poor prognosis in pancreatic cancer. In addition to being a marker of dysfunction, CD38 inhibits CD8^+^ T-cell activation via adenosine receptor signalling, which can be overcome through CD38 or adenosine receptor blockade [12]. Although blocking PD-1/PDL-1 can reinvigorate exhausted CD8 T-cells and improve control of cancer, reinvigorated CD8^+^ T-cells via anti-PD-1 ICB demonstrated minimal regeneration of an effector memory T-cell (Tem) phenotype and return to an exhausted state in the continued presence of high levels of antigen. Upon clearance of antigen, these cells exhibit an epigenetic profile distinct from either Teff-cells or Tem-cells, suggesting that while anti-PD-1 ICB transiently induces an anti-tumour effector phenotype it does so within the epigenetic landscape of an exhausted cell with little flexibility to differentiate long term in the presence of anti-PD-1 antibody [13].

In addition to targeting checkpoints, the transcription factors involved in T-cell exhaustion represent potential targets to prevent exhaustion and improve tumour killing, supported by observations that T-cells deficient in TOX are more effective at eradicating tumours [14]. The use of technologies such as CRISP-CaS9 [15] may be effective at switching off transcription factors for T-cells used in adoptive therapy, such as with CAR-T-cells, to create exhaustion resistant T-cells. Other methods, such as the inhibition of calcineurin, and subsequently, NFAT, may be an alternative method for vaccination and immunotherapies such as ICB. Such calcineurin inhibitors, typically used as immune suppressive agents, demonstrated immune potentiating functions under certain conditions [16].

**Immunosenescence.** Age-related dysregulation and decline of the immune system, termed “immunosenescence”, is associated with increased susceptibility to cancer and poor vaccine responses in older adults [17,18] due to deficiencies and dysregulation of adaptive immunity. This includes lower frequencies of naïve T-cells due to thymic involution [19], reductions in circulating CD4^+^ and CD8^+^ memory T-cells [20], lower CD28 expression and TCR responsiveness, and reduced proliferation [21,22]. These defects are paired with the accumulation of terminally differentiated T-cells with altered effector functions [23] skewed Th1/Th2 differentiation [21] reduced effector function [24] and a restructuring of lymph node architecture [18]. An investigation using aged tumour bearing mice demonstrated that T-cell dysfunction can limit the efficacy of ICB therapy due to reductions in IFN-γ expression and antigen presentation [25], consistent with the observations in human studies. In this model, innate immune priming was capable of improving ICB responses. Cancer is a disease typically effecting older people, due in part to reduced tumour surveillance. This needs to be taken into consideration prior to the use of cancer vaccination or immunotherapy given the potential for increased immune suppression [26], more frequent immune-related adverse events in relation to ICB [27] or the need to increase antigen-dose in vaccination [28].

**Chronic infection and Inflammation.** Individuals are thought to experience between 8–12 chronic viral infections throughout life [29]. These infections can have a profound effect on the function of other T-cells in a process referred to as bystander activation which occurs due to inflammation caused by infection and prolonged exposure to proinflammatory cytokines acting via multiple inflammatory pathways in the absence of antigenic stimulation. This leads to defects in CD8^+^ T-cell differentiation from effector to memory T-cells mediated via the transcription factors T-bet and Blimp-1 [30]. A study of bystander activation using a murine model of chronic viral infection observed a profound effect on total numbers, phenotype, and function of memory bystander T-cells, which severely compromised the primary expansion and effector cell formation of CD8^+^ T-cells after subsequent antigen encounter and was associated with systemically elevated levels of TNF-α and IL-6. These bystander CD8^+^ T-cells exhibited an exhausted phenotype expressing CD39, PD-1, Tim-3, CTLA-4, and downregulation of IL-7Rα, CD62L, and CD44, markers of central memory CD8^+^ T-cells. Reductions in T-cell effector cytokines were associated with increased inflammatory cytokine expression including IL-6, implicated in the STAT-1 dependent induction of exhaustion in bystander activated CD8^+^ Tem-cells, reducing their ability to produce cytokines and undergo secondary expansion upon encounter with cognate antigen [31]. Consistent with these observations are studies showing that during SARS-CoV-2 infection IL-6 is associated with impaired CD8^+^ T-cell functionality [32] whilst a dysregulated, proinflammatory signature of circulating T-cells at baseline is associated with subsequent severe COVID-19 [33]. Chronic exposure of T-cells to the inflammatory cytokine tumour necrosis factor (TNF)-α acts to uncouple the cell to TCR signal transduction pathways by impairing the assembly of the TCR-CD3 complex and suppressing calcium signalling [34]. TNF-α can also downregulate the TCR ζ-chain, a component of the TCR complex, and inhibit IL-2 cytokine expression from CD4^+^ T-cells [35]. A murine model of chronic viral infection demonstrated the induction of bystander activation of naïve CD8^+^ T-cells, resulting in altered phenotypes and reductions in number. Primary expansion of these cells upon antigen encounter may also be impaired. These observations were dependent upon the exposure of DCs to chronic infection, since DCs from uninfected mice were able to effectively prime CD8^+^ T-cells when introduced into chronically infected mice [36]. Inflammatory mediators are also capable of interrupting the resolution of T-cell activation, and thus, impede their ability to form long-lived memory cells, evidenced by the expression of an inflammatory transcriptional signature in CD8^+^ T-cells during chronic infection. However, the accumulation of T-regs is capable of resolving inflammation, initiating tissue repair and promoting the development and maturation of protective memory CD8^+^ T-cells through the inhibitory action of IL-10 on dendritic cells [37]. In line with this are observations that T-regs localised to sites of inflammation promote the generation of tissue resident memory CD8^+^ (Trm) T-cells associated with improved viral clearance and anti-tumour immunity, dependent upon the expression of TGF-β signalling in CD8^+^ T-cells [38]. IL-10 was also studied alone or in combination with Pembrolizumab, due to its anti-inflammatory properties and its ability to support cytotoxic T-cell responses. A phase I trial of AM0010, a pegylated IL-10, led to systemic immune activation with elevated immune-stimulatory cytokines and reduced TGF-β in serum alongside an objective response rate of 27% in patients with uveal melanoma and RCC. A study of AM0010 combined with pembrolizumab in 19 patients with RCC, NSCLC or Melanoma resulted in increased Th1 cytokines and proliferation of PD1+ activated CD8 T-cells while decreasing the proliferation of Tregs in peripheral blood. Tumour biopsies showed that the combination also increased the number of TIL expressing Granzymes. Objective responses were observed in 8 patients (42%) [39].

Chronic underlying inflammation, termed ‘inflammaging’, can also affect T-cell responses [40]. The aetiology of inflammaging is complex and may include factors such as genetic predisposition, obesity, gut permeability, and defects in innate immune cell populations. The resulting proinflammatory state is characterised by high circulating levels of proinflammatory cytokines including IL-1β, IL-6, IL-8, type I IFNs and TNF-α. This inflammatory environment promotes the recruitment and activation of neutrophils, macrophages, and NK-cells whilst inducing T-cell dysfunction described above, and thus, limits the ability of CTL to effectively combat viral replication or tumours, resulting in chronic antigenic stimulation and subsequent T-cell exhaustion [40]. The short-term inhibition of inflammation in older people using losmapimod, an inhibitor to Mitogen-activated protein kinase (MAPK) P38, improves antigen specific T-cell responses via inhibition of prostaglandin E2 [41]. A recent study in a murine tumour model targeted the COX-2/PGE2 pathway with widely used nonsteroidal and steroidal anti-inflammatory drugs and showed synergy with ICB, characterised by acute IFN-γ-driven transcriptional remodelling of tumours related to cytotoxic T-cell activity and indicative of efficacious ICB therapy in patients. Monotherapy with COX-2 inhibitors or PGE2 receptor antagonists rapidly induced this response program and, in combination with ICB, increased the intratumoral accumulation of effector T-cells [42]. However, in a study of patients with melanoma, ICB response rates and overall survival (OS) were similar between patients who took, or did not take, NSAID although a trend towards progression free survival (PFS) was observed (median PFS: 8.5 vs. 5.2 months; *p* = 0.054) [43]. A recent systemic review on anti-inflammatory drugs and ICB therapy found that the concomitant use of steroids was associated with improved OS and PFS whilst the use of aspirin was associated with improved PFS. However, consistent with the study detailed above, NSAIDs were not associated with clinical benefit [44]. The dose, timeframe, and treatment duration of concomitant NSAID use in these trials is unclear and it is possible that anti-inflammatory medication of a lower intensity than that of steroids will need to be received prior to, rather than merely during, ICB immunotherapy to prove efficacious. In the setting of vaccination, little is known regarding a putative effect of NSAID on vaccine mediated T-cell responses. Some evidence suggests that antibody titre can be suppressed and that NSAID can reduce the expression of type I IFN during viral infection indicating that NSAID effect aspects of immune function related to T-cells and that further study is warranted [45].

The effects of immune exhaustion, senescence and inflammation has prompted investigations intended to quantify the immune system of cancer patients prior to the administration of immunotherapy. The neutrophil-lymphocyte ratio (NLR) and systemic immune-inflammation index (SII), calculated as peripheral blood platelet counts × the ratio of neutrophil/lymphocyte counts (counts per litre) [46], are markers of immune-inflammatory status and emerged as predictors for prognosis of various tumours and their responsiveness to immunotherapy. Both high-SII and -NLR were significantly associated with a lower OS for pancreatic cancer patients treated with ICB [47]. Low-NLR and -SII are also independently associated with longer OS in patients with advanced NSCLC treated with the anti-PD-1 antibody Nivolumab [48]. Another study of NSCLC patients receiving single agent nivolumab as second line therapy showed that patients with signs of pretherapeutic inflammation including elevated NLR, SII, IL-6 and IL-8 showed significantly lower responses to ICB treatment, and reduced PFS, whereas elevated levels of IFN-γ could characterize a subgroup of patients who significantly benefit from ICB treatment [49]. In line with these observations are studies in mice associating IFN-γ-STAT-1 dependent transcriptional changes in tumours responsive to ICB, whilst transcriptomic changes associated pro-tumorigenic processes underly ICB non-responsiveness [42] consistent with observations made in patients with melanoma and nonresponsive to ICB [50]. Other immune based markers of immunotherapy responsiveness include the tumour mutational burden and the presence of TIL used to generate an ‘Immunoscore’ based upon CD8^+^ and memory CD45RO+ T cells currently under investigation [51].

Collectively studies on T-cell exhaustion, immune senescence and chronic inflammation implicate chronic antigenic stimulation and signalling by inflammatory cytokines in altering the transcriptional profile and subsequent function of activated T-cells. This results in poor effector function and impediments to the generation of memory, imbalances in CD4^+^ T-cell cytokine expression, and the onset of exhausted phenotypes expressing checkpoints (Figure 1). In the setting of cancer these T-cells also face the induction of immune suppression within the TME characterised by hypoxia, acidosis, metabolic stress, and inhibition by immune suppressive cells. This sequence of events disproportionately impacts individuals in which T-cell immunity is already contracting through the effects of age. Therefore, methods to reset or prime T-cell immunity prior to vaccination or immunotherapy and to subsequently maintain these responses have the potential to significantly improve patient outcomes.

## 3. Directing Anti-Tumour T-Cell Responses

Directing T-cell responses for vaccination or immunotherapy can be defined as the use of immune modulatory agents to prime T-cells either to promote effector responses or support their differentiation into long lived memory T-cells. The type of priming required depends upon the stage of T-cell differentiation [52], which is in turn controlled by T-cell metabolism [53]. Naïve T-cells maintain a state of hypo-responsiveness, with low proliferation and metabolic activity prior to differentiation into effector T-cells upon activation in lymph nodes by DC presenting cognate antigen. These Teff-cells dramatically increased glucose and amino acid metabolic requirements, which may significantly deplete amino acid availability to other organs [54], and preferentially utilise aerobic glycolysis supporting the release cytotoxic granules and effector cytokines upon engagement of their TCR with tumour cells. Memory T-cells are generated from the Teff-cell population depending upon the nature of TCR signalling, co-stimulation and use of fatty acid oxidation (FAO) and oxidative phosphorylation (OXPHOS)-based metabolism. Memory cells can be divided into three subsets. Effector memory T-cells (Tem-cells) have a higher effector function compared to central memory T-cells (Tcm-cells), which instead demonstrate greater persistence and anti-tumour immunity with the Tcm/Teff ratio a predictive biomarker of immune responses against some tumours [55]. Another memory subtype, T stem cell memory, have increased proliferative potential and can reside in dense antigen-presenting-cell niches within tumours, where they proliferate to form terminally differentiated T-cells with effector function [56]. Importantly, T-cell modulatory agents targeting T-cell signalling or metabolism may have little anti-tumour efficacy when used as single agents but can support the three signals received from naïve T-cells during their first encounter with antigen or support therapeutic vaccination, the use of ICB or adoptive T-cell therapy, which typically aim to reactivate existing anti-tumour immunity, but often fail when T-cells are not primed to respond. Directing T-cells to reach an ‘immune set point’ optimal for T-cell based vaccination or immunotherapy may involve a number of different approaches including the inhibition of chronic inflammation, costimulation through the use of cytokines or agonist antibodies, modulation of the gut microbiome, dietary or bacterial metabolites, small molecule modulators of TCR signalling pathways, or the modulation of cellular metabolism. These agents act against either naïve, effector or Memory T-cell subsets through the alteration of TCR signalling, costimulation, transcriptional regulation, and the modulation of mitochondrial biogenesis and metabolism (Figure 2). The most promising agents and their mechanisms of action will be discussed here.

## 4. T-Cell Co-Stimulation

T-cell activation is a complex process involving the reception and interpretation of multiple extra and intracellular signals in addition to TCR based signal transduction. Both cytokine signaling and co stimulatory receptors shape the extent and nature of T-cell activation and are under investigation as cancer immunotherapies.

**T-cell Cytokines:** Activation of T-cells results in significantly increased metabolic demands whilst the migration into TME exposes anti-tumour T-cells to competition with tumour cells for metabolic resources. Methods to program T-cell metabolism to generate more robust T-cells holds great promise for T-cell vaccination, the use of immunotherapies such as ICB or adoptive cell therapy with expanded TIL or T-cells engineered ex vivo with CARs or TCRs. T-cell stimulatory common γ chain cytokines IL-7, IL-15, or IL-21 are capable of supporting the generation of memory cells with improved mitochondrial fitness and less overt T-cell differentiation compared to the use of IL-2, improving anti-tumour immunity [57,58,59]. For example, IL-7 and IL-15 are capable of increasing the expression of aquaporin 9 [60] and Carnitine palmitoyltransferase (CPT)-1a [58] inducing mitochondrial biogenesis and metabolically reprograming T-cells to utilise fatty acid esterification, triglyceride (TAG) synthesis and FAO thus providing the metabolic changes to energy production necessary to promote a memory phenotype.

IL-7 signals via IL-7Rα complexed with the common-γ chain and is essential to the development of T-cells. IL-7Rα is highly expressed on naïve T-cells, and memory T-cells in the periphery [61]. In a murine tumour model the addition of IL-7 alone had no effect, however adjuvant IL-7 improved a vaccine induced anti-tumour response. In this study IL-7 induced CTLs with higher expression of granzyme-B, faster, amplified degranulation kinetics and more effective cytolytic activity. IL-7 signalling in T-cells was able to repress the expression of casitas B-lineage lymphoma b (Cbl-b), a negative regulator of T-cell activation [62]. An autologous tumour vaccine modified with recombinant new castle disease virus to express IL-7 promotes anti-tumour immune responses in murine models by increasing the IFN-γ production and cytotoxicity of CD8^+^ T-cells [63]. IL-7 can also expand T-cells with low-affinity TCRs [64]. In humans, administration of recombinant human IL-7 increases in vivo TCR repertoire diversity by preferentially expanding naïve T-cell subsets without increasing T-regs but doesn’t enhance T-cell cytokine expression [65]. A phase II trial of rhIL-7 in lymphopenic metastatic breast cancer patients has demonstrated a significantly increased CD4^+^ and CD8^+^ T-cell count [66]. These studies indicate that IL-7 is capable of increasing T-cell proliferation without the toxicity associated with IL-2 but is ineffective in promoting anti-tumour immunity as a single agent.

IL-15 demonstrated efficacy in murine models of cancer [67] whilst studies in Macaques have demonstrated large increases in Tem CD8^+^ T-cell numbers [68], providing a rationale for its use as an alternative to IL-2. However, IL-15 administered as monotherapy has proven ineffective, with increases in CD8^+^ T-cells but no objective responses [69] suggesting that IL-15 may be better suited in combination with other immunotherapies. An issue with the use of IL-15 is its poor pharmacokinetics. This prompted the development of various IL-15 agonists [70] under investigation in clinical studies. For example, ALT-803 is an IL-15-IL-15Rα fusion protein fused to an IgG-Fc fragment to increase half-life in vivo. A combined of ALT-803 with anti-PD-L1 demonstrated improved anti-tumour control in murine model [71], prompting a clinical trial of 21 NSCLC patients in which 6 (29%) showed a partial response, and a further 10 demonstrated stable disease; the median overall survival was 17.4 months [72].

IL-21 is also under investigation as an immunotherapy. IL-21 promotes the generation of murine Teff-cells from naïve precursors, enhancing expansion and supporting the development of cytotoxicity, albeit with reduced IFN-γ expression [73]. IL-21 can also rescue T-cells from exhaustion [74], promote T-cell survival [75] and increase the expression of granzyme -B from CTL [76]. A vaccine construct consisting of OVA antigen and +/− IL-21 induced greater CTL proliferation and effector cytokine IFN-*γ* expression with IL-21 in OVA expressing tumours in mice via a PI3K and mTORC1 pathway [77]. The fusion of IL-21 to anti-PD-1 antibody demonstrated potent anti-tumour effects in established tumour-bearing mice, accompanied with an increased frequency of Tscm-cells and expansion of tumour-specific CD8^+^ Tem-cells which was superior to a combination of PD-1 blockade and IL-21 infusion [78]. In humans, a phase II trial of IL-21 monotherapy as first line treatment for patients with limited-disease metastatic melanoma demonstrated 25% OR with 9 partial responses, a PFS of 4.3 months and median OS of 12.4 months indicating that IL-21 has anti-tumour effects as a single agent [79].

Combinations of these cytokines are being investigated in preclinical studies. Vaccination of plasmids expressing IL-21 and IL-15 promoted the expansion of CD8^+^ memory T-cells and enhanced CD8^+^ T-cell responses against viral antigens, which was partially independent on CD4^+^ T-cell help [80], whilst coexpression of IL-21 and IL-7 in a whole-cell cancer vaccine enhanced anti-tumour immunity dependent upon both CD4^+^ and CD8^+^ T-cells capable of generating effector memory cells associated with tumour infiltration and long-term anti-tumour function in murine models [81]. Alternatives to IL-2 based immunotherapy are in development. These include Bempegaldesleukin (BEMPEG/NKTR-214), a CD122-preferential IL-2 pathway agonist demonstrating promising anti-tumour efficacy in pre-clinical studies of advanced cancer in combination with ICB and local radiation therapy [82]. A single arm phase I trial of NKTR-214 plus nivolumab in 38 patients with selected immunotherapy-naïve advanced solid tumours demonstrated a 59.5% OR including 7 CR and increased infiltration, activation, and cytotoxicity of CD8^+^ T-cells without promoting T-reg subsets. At higher doses clinical effects were seen irrespective of PDL-1 status or baseline TIL indicating that the combination might be effective in patients who typically do not respond to anti-PD-1 monotherapy [83].

Non-Cγ cytokines being studied as immunotherapies include interleukin-12, which is produced by myeloid cells such as DC and provides costimulatory signals to T-cells. An IL-12 gene expressing plasmid immunotherapy called GEN-1, formulated with PEG-PEI-cholesterol lipopolymer and delivered intraperitoneally at the tumour site in advanced ovarian cancer to stimulate immune responses and limit toxicity [84]. GEN-1 is being studied in patients with ovarian cancer alongside chemotherapy (NCT03393884). Other IL-12 family cytokines are under investigation for potential immunotherapeutic properties [85].

The studies detailed here identify Cγ chain cytokines and constructs as exciting agents for use in combination with both vaccines and ICB based immunotherapy. The ability of these cytokines to promote both effector T-cells and the generation of memory indicate that they have the potential to be versatile agents used in multiple settings (Table 2). **T-cell agonists:** Full T-cell activation requires co-stimulation and T-cells express numerous receptors capable of providing these signals upon ligation with receptors on antigen presenting cells. CD28, a member of the immunoglobulin superfamily (IgSF), is constitutively expressed on the cell surface of naive T-cells and localises with the TCR in the central supra-molecular activation complex. CD28 provides an essential costimulatory signal for T-cell growth and survival upon ligation by B7-1 which is constitutively expressed on APC and B7-2 which is expressed upon maturation of APC [86]. Numerous other costimulatory or coinhibitory receptors were discovered including IgSF immune checkpoints PD-1 (CD279), CTLA-4 (CD152), TIGIT, TIM-3 and LAG-3 (CD224) [87] whilst others, particularly type-V receptors of the tumour necrosis factor receptor superfamily (TNFRSF), are involved in costimulation including OX40 (CD134), 4-1BB (CD137), Glucocorticoid-induced tumour necrosis factor receptor (GITR/CD357) and CD40L. The direct activation of these costimulatory pathways using agonistic antibodies, alone or in combination with other immunotherapeutic strategies, is under investigation.

**4-1BB.** 4-1BB is expressed on activated CD4^+^ and CD8^+^ T-cells, T-regs and NK-cells whilst its ligand, 4-1BBL, is expressed on mature DCs. On CD8^+^ T-cells 4-1BB expression peaks early after TCR stimulation and upon binding to 4-1BBL interacts with TRAF1 and TRAF2 which subsequently activates MAPK and JNK signalling pathways, resulting in increased cytokine expression and prolonged survival. In preclinical studies using a colorectal cancer mouse model, a 4-1BB agonist demonstrated dose-dependent suppression of tumour growth [88]. A human engineered IgG2 antibody against 4-1BB, utomilumab was studied in a phase 1 trial of 55 patients with advanced solid cancers and demonstrated an ORR of only 3.8%. This study indicated that utomilumab was safe and had potential anti-tumour activity with levels of circulating lymphocytes a potential biomarker predicting clinical benefit [89]. Trials of the anti-4-1BB antibody urelumab, demonstrated increased levels of the activation of peripheral CD8^+^ T-cells and IFN-inducible genes [90] but were hampered by a high incidence of toxicity at higher doses (1 to 5 mg/kg). However, lower doses of 0.1 mg/kg are considered safe, and are associated with immunologic activity demonstrated by the induction of IFN-inducible genes and cytokines [91].

Murine models support the use of 4-1BB agonists in combination with anti-PD-1 ICB in promoting effective anti-tumour immune responses, demonstrating an elevated CD8^+^/Treg ratio and increased activity of tumour-specific cytotoxic T lymphocytes and enrichment of genes including IFN-γ, and EOMES [92]. In a mouse ovarian carcinoma model anti-4-1BB and anti-PD-1 administration improved survival, increased effector CD8^+^ T-cell density, and decreased the number of T-regs and MDSC [93]. In humans, a phase 1b study of utomilumab in combination with pembrolizumab for the treatment of advanced solid tumours was well tolerated and demonstrated an ORR of 26% including one complete remission [94]. A phase 1/2 study demonstrated the safety of the combination of urelumab and nivolumab in patients with advanced malignancies. The objective response rate was 10.5% (9 of 86 patients). In this study, adding nivolumab to urelumab did not produce substantial additive/synergistic benefits at the evaluated dose levels.

**OX40.** OX40 is expressed on CD4^+^ and CD8^+^ T-cells upon activation and is present for about 4 days thereafter. Binding of OX40L to OX40 promotes T-cell expansion and survival via enhanced expression of IL-2 and antiapoptotic proteins such as BCL-2, promotes the generation of memory T-cells and the inhibition of T-regs [95]. OX40 antibody agonists in combination with anti-TGF-β antibodies suppresses T-reg function and increases the expression of IFN-γ [96], however, OX40 can also induce the proliferation of T-regs, suggesting it modulates T-regs in numerous ways based upon the nature of the immunological environment [97]. The use of a murine model of anti-PD-1–refractory mammary cancer was used to study the combination of anti-OX40 with anti-PD-1 and demonstrated that concomitant use anti-PD-1 antibody attenuated the therapeutic effects of anti-OX40 including increases in exhaustion markers and significantly reduce CD4^+^ and CD8^+^ T-cell proliferation compared to untreated mice. In contrast the use of anti-PD-1 ICB after anti-OX40 antibodies improved on the efficacy of single agents. Thus, the combination of agonist and ICB may depend upon the sequence [98].

A phase I trial of an OX40 agonist in 30 patients with advanced solid tumours showed potential efficacy associated with increased proliferation of both CD4^+^ and CD8^+^ T-cells [99]. Another phase I trial of the humanized anti-OX40 agonist MEDI0562 as single agent therapy in patients with solid tumours demonstrated increases in CD4^+^ and CD8^+^ memory T-cell proliferation and decreased intratumoral OX40+ FOXP3+ cells, 44% of patients demonstrated stable disease [100]. A phase 1 trial of 44 patients administered the OX40 agonist MOXR0916 combined with an anti-PDL-1 ICB demonstrated only 2 responses [101]. Another phase I trial, of OX40 agonist PF-04518600 against hepatocellular carcinoma, demonstrated safety, whilst 63% remained progression free after treatment [102]. Other trials of OX40 agonists, such as BMS-986178 given with or without nivolumab or ipilimumab (NCT02737475) and GSK-3174998 given alone or in combination with pembrolizumab (NCT02528357), are ongoing and results are pending.

**Glucocorticoid-induced TNFR-related protein.** GITR is highly expressed on T-regs and present at lower levels on both naïve and memory T-cells. GITR enhances T-lymphocyte activity after suboptimal TCR stimulation by upregulating IL-2 and IFN-γ and enhances T-cell survival by inhibiting TCR activation induced apoptosis. The agonist anti-GITR antibody DTA-1 induced regression of small established B16 melanoma tumours by impairing the accumulation of T-regs in the TME through downregulating foxp3 expression and enhanced tumour-specific CD8^+^ T-cell activity [103]. A phase 1 study of the GITR agonist AMG228 involving 30 patients with advanced solid tumours demonstrated stable disease in 7 patients; however, T-cell activation and anti-tumour activity was not correlated with GITR expression [104]. A GITRL/IgG1 agonist fusion protein, MEDI1873, investigated in a phase 1 study of 40 patients with advanced tumours demonstrated stable disease in 42.5% of patients. MEDI1873 induced a 25% decrease in GITR+/FOXP3+ T-cells and increased markers of proliferation on CD4^+^ T-cells at higher doses [105].

In a preclinical study using a murine ovarian cancer model, the combination of a GITR agonist and a PD-1 inhibitor reduced peritoneal tumour growth by increasing the responses of memory T-cells cells against tumour cells, increased frequencies of IFN-γ-producing effector T-cells and suppressed Tregs and MDSC [106]. A study of MK-1248 alone and in combination with pembrolizumab in 17 patients advanced solid tumours demonstrated one CR and two had a PR in the combination group with an ORR of 18% [107]. A phase I study of anti-GITR antibody MK-4166, as monotherapy or with pembrolizumab in patients with advanced solid tumours, demonstrated an ORR of only 2.2%; however, in a melanoma expansion group, 8 out of 13 patients with immune checkpoint inhibitor (ICB)-naïve melanoma achieved a response for an ORR of 62% [108].

**Inducible T-cell Co-stimulator**. ICOS is related to CD28 and its expression is induced upon T-cell activation. Ligation of ICOSL on DC can induce proliferation and cytokine expression in T-cells whilst suppressing T-reg function through the induction of Foxp3. In murine models, a humanised ICOS agonist JTX2001, in combination with anti-PDL-1 ICB, demonstrated anti-tumour activity associated with ICOS expression in TME resident T-regs [109]. A subsequent trial of JTX2011 in combination with Nivolumab demonstrated a PR in 7.5% of patients [110]. Another trial, testing the ICOS agonist GSK3359609, used as a monotherapy or in combination with anti-PD-1 ICB with pembrolizumab in patients with head and neck squamous cell carcinoma also demonstrated modest response rates with and ORR of 8% for the monotherapy group and 28% for the combination group [111].

**CD40-CD40L.** CD40 ligand is another TNFSFR present on activated T-cells which can support T-cell activation upon ligation of CD40 present on APCs such as DC. CD40/CD40L can induce the maturation of DC, increasing the expression of co-stimulatory proteins such as CD80, and allow DC to activate CD8^+^ T-cells in the absence of T-cell help. In a melanoma mouse model, an adenovirus encoding CD40L was administered intratumourally in combination with anti-PD-1 ICB. This led to the generation of tumour-specific, CD8^+^ T cells and systemic tumour eradication, which notably included both the primary subcutaneous tumour and brain metastases, indicative of an immune abscopal effect [112]. Another murine study, using an anti-CD40 agonist, found that increased tumour density was vital to its efficacy, which was dependent on CD8^+^ T-cells and independent of CD4^+^ T-cell help. The requirement for inoculation with relatively large tumours was hypothesized to be due to the need to provide antigen to CD40 agonist activated DC [113]. The ability of CD40 to promote DC maturation and presentation of tumour antigen indicates that CD40 based immunotherapies may be effective in combination with vaccination or radiotherapy, capable of inducing immunogenic tumour cell death, which is currently under investigation in a clinical trial (NCT03165994).

These promising preclinical studies led to numerous clinical trials utilising soluble CD40L or CD40 agonistic antibodies, which demonstrated modest anti-tumour responses thus far [114]. One of these, the agonistic CD40 antibody CP-870893 (selicrelumab), has demonstrated a maximum tolerated dose of up to 0.2 mg/kg body weight and, combined with anti-CTLA-4 ICB, showed promising responses and long-term survival in a minority of patients who subsequently received other therapies such as anti PD-1 ICB [115]. Anti-CD40 agonist antibodies need to cross link to effectively activate CD40 signalling. The Agonist antibody APX005M utilizes an IgG1 domain to promote binding to Fc receptors and thus promote cross linking. This candidate therapeutic is currently being tested in a number of clinical trials including as a combination partner with anti-PD-1 ICB and a personalized cancer vaccine NEO-PV-01, consisting of up to 20 neoantigen peptides with adjuvanticity using Poly-ICLC, in patients with advanced melanoma (NCT03597282). The use of CD40 agonists is an attractive approach, particularly with immunotherapy approaches involving the use of tumour antigen. However, like the other agonists described here, CD40 agonists may suffer from significant toxicity, especially in combination with ICB. Whilst other agonists targeting TNFRSF members directly stimulate T-cells, CD40 agonists activate APC such as DC. The targeting of both DC and T-cells through combined therapy has the potential to act on multiple stages of the anti-tumour immune response and many other innate immune agonists are being studied to therapeutically activate DC [116].

The studies of T-cell costimulatory receptor agonists show that the use of single agonists is largely ineffective with objective responses often less than 10%. However, the use of urelumab or MK-4166 in combination with anti-PD-1 ICB demonstrated promising ORR (Table 2). Issues with the use of T-cell agonists include the induction of T-cell exhaustion, toxicity and the optimal sequencing of agonist and ICB. The use of agonist treatment prior to ICB and/or in combination with treatments capable of inducing epitope spreading such as radiotherapy [117] or oncolytic therapy [118] may be advantageous since the costimulation mediated by the agonists would likely act on the expansion phase of naïve to effector T-cells. This could be followed by subsequent ICB therapy to ameliorate exhaustion.

**Table 2 vaccines-09-01392-t002:** T-cell costimulatory agents under investigation as cancer immunotherapies.

Co Stimulation	Effect on T-Cells	Questions	Ref
IL-7	Mitochondrial biogenesis metabolically reprograming T-cells to utilise fatty acid oxidation. improves the vaccine induced anti-tumour response and survival. Increasing the IFN-γ production and cytotoxicity of CD8^+^ T-cells	IL-7 may not have therapeutic efficacy when used alone and thus requires consideration regarding which immunotherapies may be amplified by IL-7.	[63,64,65]
IL-15	Promising responses in combination with anti-PD-1 therapy in early clinical trial	IL-15 administered as monotherapy proved ineffective with increases in CD8^+^ T-cells but no objective responses. poor pharmacokinetics need to be addressed.	[69,70,72]
IL-21	Promote the generation of effector T-cells from naïve precursors, enhancing expansion and supporting the development of cytotoxic effector function, may demonstrate efficacy as a single agent	Proliferative responses to IL-21 may be associated with reduced IFN-γ expression. Not yet studied in clinical trials in combination with ICB	[73,79]
NKTR	Can induce durable anti-tumour T-cell responses and demonstrates efficacy in poorly immunogenic tumours	Efficacy needs to be demonstrated in a multiarmed trial which is underway	[82,83]
Anti-1-44B	Increased cytokine expression and prolonged survival, suppression of tumour growth	Low response rate and potential toxicity need to be addressed	[87,88]
Anti-OX40	Promotes T-cell expansion and survival via enhanced expression of IL-2 and anti-apoptotic proteins such as BCL-2 and promotes the generation of memory T-cells and inhibition of T-regs	Low objective response rate in clinical trials. The need to carefully time administration of OX40 agonist before ICB therapy.	[93,94,95,96,97,98,99]
Anti-GITR	Enhances T-lymphocyte activity after suboptimal TCR stimulation by upregulating IL-2 and IFN-γ and enhances T-cell survival by inhibiting TCR activation induced apoptosis.	Low response rate demonstrated in early trials	[101,103,106]
Anti-ICOS	Proliferation and cytokine expression in T-cells whilst suppressing T-reg function	A modest patient response in clinical trials thus far	[107,108]

## 5. The Gut Microbiome and Dietary Interventions

The gut microbiome consists of approximately 1 × 10^13^ bacteria comprised from over 1,000 species with about 160 species present per individual [119]. The gut microbiome weighs about 250 g and expresses approximately 3 million different genes, representing a coding capacity 150-times larger than that of the human genome. A number of studies between 2014–2019 revealed the essential role of the gut microbiome in calibrating T-cell immune responses in the context of chemotherapy and cancer immunotherapy [120,121,122,123,124,125]. These studies demonstrated that particular microbiomes, and occasionally specific microbes, were associated with the efficacy of chemotherapies such as cyclophosphamide or immunotherapies such as ICB in solid tumours including melanoma or NSCLC. Remarkably, faecal microbiota transplant (FMT) from ICB responder patients to non-responder patients proved capable of re-capitulating the responder phenotype in the previously nonresponsive patients and is associated with increased CD8^+^ T-cell infiltration of tumours [126,127]. Additional evidence supporting the role of the gut microbiome comes from observations that oral antibiotic use may hinder ICB efficacy [128], antibiotic-induced perturbation of the gut microbiota significantly increases tumour progression in multiple breast cancer murine models [129], whilst germ-free mice fail to develop effective memory CD8^+^ T-cell responses [130]. Recent studies have extended these findings to associate patient diet [131] or probiotic consortia [132] in ICB responsiveness.

A surprising observation was that the bacteria associated with ICB responses differed between each study prompting the view that communities of microbes are more important than the presence of particular species, consistent with the use of FMT or a defined bacterial consortium in supporting ICB licenced, CD8^+^ T-cell dependent, anti-tumour immune responses [132]. These basic studies began to be translated into clinical research using specific microbes or consortia of microbes as probiotic immunotherapeutics. This bacterial consortium, VE800, is now undergoing clinical trial in combination with nivolumab in during which VE800 will be taken daily after 5 days of vancomycin treatment (NCT04208958). The use of consortia or FMT is supported by observations that microbiomes with relatively high diversity are associated ICB efficacy and increased Th1 and CD8^+^ T-cell activation whilst a >1.5 ratio of beneficial bacteria to nonbeneficial bacteria was found to predict anti-PD-1 ICB efficacy in metastatic melanoma [122].

However, most probiotics involve the use of a single bacterial species. EDP1503 is a monoclonal strain of Bifidobacterium animalis lactis capable of inducing systemic anti-tumour immunity by activating both innate and adaptive responses characterised by increased cytokine expression including IFN-γ and CXCL10, activation of CD8^+^ T-cells and a proinflammatory signature within the TME [133]. A combination of EDP1503 and pembrolizumab was studied in a clinical trial (NCT03775850) which showed treatment with single-agent EDP1503 2 weeks prior to combination dosing with pembrolizumab led to a PR in 2 patients with triple-negative breast cancer (TNBC) and 2 with NSCLC representing an ORR of 14% across 29 patients [134]. Microbial Ecosystem Therapeutics (METs) are consortia of human-derived bacteria designed to be reproducible, scalable, and safe alternatives to faecal transplant. MET4 is a consortium of taxa associated with ICB responsiveness. In a phase I trial of MET4 in combination with ICB greater number of MET4-associated taxa were detectable in MET4 recipients than controls. No significant change in Shannon diversity after MET4 was observed, however controls were more likely to lose diversity overtime than MET4 recipients and *Bifidobacterium*, *Collinsella,* and *Enterococcus* were significantly more common and abundant in MET4 recipients than controls [135]. The microbiome restoration biotherapeutic, MaaT013, consists of a standardised, pooled FMT product of approximately 455 bacterial species compared to 274 on average for FMT. MaaT013 enemas will be administered to patients with unresectable or metastatic melanoma in combination with ipilimumab and nivolumab and compared to ipilimumab and nivolumab without MaaT013. This study will also measure plasma levels of IL-6, IL-8, MCP1, IL-1β, TNF α, sCD25, sCTLA-4, sPD-L1 and SCFA (NCT04988841). *Enterococcus gallinarum*, a commensal Gram-positive bacterium, is associated with ICB responsiveness [120]. MRX0518 is a strain of *E. gallinarum* and monotherapy with MRX0518 was able to reduce tumour size in syngeneic mouse models of breast and lung carcinoma associated with an increase in the CD8^+^ T-cell: Treg ratio. In vitro MRx0518 induce cytokine production and immune cell activation [136], which appears to be due, in part, to the immune stimulatory properties of the bacteria’s flagellin [137]. A clinical trial of MRx0518 and Pembrolizumab to treat patients with advanced solid tumours having progressed on anti-PD-1/PDL-1 monotherapy is ongoing (NCT03637803). *Clostridium butyricum* MIYAIRI 588 (CBM 588) is a probiotic bacterium studied to prevent or ameliorate dysbiosis in the gut. *C butyricum* is a Gram-positive bacillus which produces short-chain fatty acids and can promotes the growth of commensal gut bacteria such as *Bifidobacterium*, *Lactobacillus* and *Bacteroides* species, and was shown to enhance Th1 cells and IL-10 production. A phase I trial to assess the biologic effect of CBM588 in combination with nivolumab plus ipilimumab in patients with metastatic renal cell carcinoma is underway (NCT03829111) [138]. A probiotic cocktail of bacteria associated with ICB responsiveness, SER-401, was studied in a Phase 1b clinical trial combined with Nivolumab in ICB naïve melanoma patients. In this study the SER-401-Nivolumab combination is administered after gut microbiome depletion using the antibiotic Vancomycin. However, this trial was discontinued during enrolment (NCT03817125). Strategies to modulate the gut microbiome for example using high fibre diets [139] or ginger extract in the setting of cancer are also underway (NCT03268655), whilst another study underway aims to test the administration of dietary supplementation with resistant starch to modify gut microbiomes in patients undergoing treatment for solid cancers with combined ICB (NCT04552418).

These metabolites include short chain fatty acids (SCFA), bile acids, and amino acid metabolites such as putrescine, spermidine, spermine, phenolic, and indolic compounds [140]. SCFA including butyrate, acetate and propionate are amongst the main metabolites produced by gut resident microbiomes. These SCFA act locally to regulate gut immunity through the induction of T-cell tolerance and activation of T-regs. SCFA also act systemically on T-cells to promote ant-tumour immunity. Butyrate signalling through G protein coupled receptor (GPR)41 and GPR43 can increase the memory potential of effector CD8^+^ T-cells by enhancing metabolism and uncoupling the Krebs cycle from glycolytic input. This results in preferential fuelling of oxidative phosphorylation through sustained glutamine utilization and fatty acid catabolism, ensuring that effector CD8^+^ T-cells contract in a manner that supports a pool of circulating memory cells with the necessary metabolic adaptations for long-term survival [130]. These CD8^+^ T-cells demonstrate enhanced utilisation of carbon from other sources, including SCFA themselves. Butyrate can also increase the expression of IFN-γ through its HDAC inhibitory properties and the activation of mTOR [141] indicating that butyrate can support both effector responses and their subsequent survival and transition into memory subsets. Butyrate also induces constitutive IL-10 expression from T-cells whilst the expression of IFN-γ or IL-17 induction is dependent upon specific activation. As such butyrate can promote CD8^+^ T-cell immune responses whilst providing the negative feedback allowing for the correct resolution of T-cell immunity [141]. Levels of SCFA also correlate positively with anti-PD-1 responses against solid cancers [142]. A recent study testing oral administration of powder from the fruit Ziziphus jujube to mice with colon adenocarcinoma showed alteration of the gut microbiota associated with increased production of SCFA including butyrate, increases in TIL and improved responses to anti-PD-1 ICB [143]. This approach is supported by a study using pectin to improve anti-PD-1 treatment in tumour bearing mice administered FMT from CRC patients, including those resistant to anti-PD1 ICB. In this study the addition of pectin altered the microbiome, increasing microbial diversity and enriching butyrate-producing bacteria. Pectin significantly enhanced anti-PD-1 mAb efficacy associated with increases in TIL. Administration of butyrate was sufficient to replicate the effects of pectin on anti-PD-1 efficacy in this model [144]. Further evidence that dietary fibre effects cancer immunotherapy through the generation of SCFA comes from observations that the generation of Propionate is associated with intratumoural immune responses and the efficacy of ICB in lung cancer patients taking a high fibre diet (139). Mice fed the prebiotics inulin or mucin also demonstrated changes in gut microbes, anti-tumour immune responses, and inhibition of BRAF mutant melanoma growth. Inulin could enhance the efficacy of a MEK inhibition against melanoma, whilst delaying the onset of resistance, suggesting that dietary interventions targeting the gut microbiome have the potential to combine with targeted therapy and immunotherapy [145].

Other compounds metabolised or synthesised by the microbiome were identified, some are positively associated with anti-tumour immune modulation, whilst others are decreased in cancer patients compared to healthy controls. Linoleic acid can enhance T-cell proliferation whilst decreasing the expression of inflammatory cytokines IL-6 and TNF-α [146]. Niacin, whose metabolite NAD+ discussed below, is altered in cancer patients. Microbial derived inosine is associated with the efficacy of ICB therapy in mouse models through its ability to enhance the activation of CD8^+^ T-cells via increases in the transcription factor T-bet and dependent on T-cell expression of the adenosine A2A receptor and co-stimulation [147]. Some metabolites produced by intestinal bacteria can inhibit T-cell based immune responses. These include the bacterial metabolism of tyrosine to form p-cresyl sulfate which reduces the percentage of IFN-γ-producing Th1 cells and Th1-driven delayed-type hypersensitivity (DTH) reaction in tyrosine fed mice [148]. Antibiotic induced disruption of the microbiome was shown to alter immune responses through differential expression of metabolites [149], whilst a combination of immunological and clinical parameters including soluble immune-checkpoints, performance status, and gut metabolites was assessed to identify patients more suitable for Nivolumab treatment [150].

The ketogenic diet, characterised by intermittent scheduling of fasting, has recently been associated with T-cell dependent inhibition of aggressive tumour growth in a murine model via increases in 3 hydroxybutyrate (3HB). Oral supplementation of 3HB was able to restore the effectiveness of anti-PD-1 ICB via the GPR109A-dependent inhibition of PDL-1 and expansion of CXCR3+ T-cells in this model. Ketogenic diet-derived 3HB was associated with increases in particular microbial species in the gut [151]. In humans, clinical study is underway based upon evaluating the tolerance of one year of ketogenic diet and vitamin supplementation in patients treated for a metastatic renal cell carcinoma with standard of care treatment, including ICB (NCT04316520). Diet is the focus of another clinical trial assessing whether restricted feeding to an 8–10 h window followed by prolonged nightly fasting, monitored by the use of an app, can lead to better response rates to immunotherapy in head and neck cancer patients (NCT05083416). An older clinical trial in patients with head and neck cancer tested a nutritional supplement containing vitamin D, niacin, omega-3-fatty acids, arginine, dietary nucleotides, and soluble fibre for its immune potentiating effects. The supplement was taken after undergoing postoperative radio-chemotherapy prior to the advent of immunotherapy (NCT00559156). Such nutritional support possibly renewed potential in the setting of ICB. A phase II trial will investigate the possible immune effects of diets differing in fibre content on the structure and function of the gut microbiome in patients with stage III–IV melanoma and who are being treated with pembrolizumab or nivolumab ICB (NCT04645680). Another trial in Melanoma patients taking anti-PD-1 ICB seeks to assess a high-fibre diet, recorded using an app, and least 150 min of moderate or 75 min of high-intensity exercise per week, measured using a physical activity tracker and compared to that of a non-intervention control group. The study will measure adherence, changes in the gut microbiome and effects on PFS and OS (NCT04866810).

Dietary interventions can modulate the gut microbiome but do not necessarily induce a permanent compositional shift. As such, they may have less dramatic effects compared to the use of FMT or probiotics; however, the safety and convenience of such approaches are attractive. The direct use of metabolites might represent a combination of safety, convenience, and efficacy.

Diet involving the restriction of the amino acid methionine is being studied for its potential to enhance the effects of chemotherapy in methionine dependent tumours [152]. Interestingly methionine restriction (MR) also reduces histone H3K4 methylation in CD4^+^ T-cells altering the pattern of gene expression involved in Th17 cell proliferation and cytokine production [153]. MR is implicated in the development of hyperhomocysteinemia [154] and homocysteine was shown to activate T-cells by enhancing endoplasmic reticulum-mitochondria coupling and increasing mitochondrial ROS production, calcium content, membrane potential, mass, and mitochondrial respiration, with both OXPHOS and aerobic glycolysis in T-cells were upregulated with homocysteine treatment [155]. However, a recent study indicates that MR has a negative effect on T-cell function and reduces efficacy of anti-tumour immunotherapy in immunocompetent mice associated with decreased CD8^+^ T-cell numbers and cytotoxicity. This occurs through the modulation of the gut microbiota, including reductions in *A. muciniphila*, which is implicated in anti-tumour immunity, and resulting in decreased production of hydrogen sulfide, an endogenous potentiator of T-cell activation [156]. The gut is the major H2S-producing organ, where the gut microbiota is responsible for about half of this production [157] and the inhibitory effect of methionine restriction could be replicated using FMT from methionine restricted mice and ameliorated by the supplementation of a hydrogen sulfide donor, GYY4137, which could rescue MR diet-induced resistance to anti-PD1 immune therapy in tumour bearing mice. The supplementation of L-methionine could also increase the lifespan of circulating CD8^+^ T-cells, reduce markers of exhaustion on CD4^+^ T-cells, and reduce T-regs in peripheral blood. Methionine-supplemented diet suppressed the growth of allograft CT26 tumours in immunocompetent mice but markedly enhanced CT26 tumour growth in immunocompromised mice [156].

The studies detailed here (summarised in Table 3) are investigating how best to use knowledge of the gut microbiome alongside diet and exercise to modulate immune function in cancer patients undergoing immunotherapy. They are based upon a growing understanding of the dynamic interactions between the immune system, our lifestyles, and our environment, and may represent a turning point in how patients take part in their own therapy, not just to mitigate its toxicities but to improve its efficacy.

**TCR signal transduction.** Signals transmitted via the TCR determine the extent and nature of the ensuing T-cell response. TCR signalling is thus a promising target for modulated anti-tumour immunity (selected agents are summarised in Table 4). For example, TCR signalling pathways are regulated by tyrosine phosphatases, which are under investigation as targets to improve T-cell responses against cancer. The protein tyrosine phosphatase non-receptor type 22 (PTPN22) is a negative regulator of Src and Syk family kinases downstream of the TCR and regulates effector and memory T-cell responses [158]. CD8^+^ T-cells lacking PTPN22 demonstrate improved anti-tumour immunity and resistance to suppression mediated by TGF-β, via increased IL-2 expression [159]. Responses to weak affinity tumour antigens are also enhanced by knockout of PTPN22 [160] whilst PTPN2 deficient T-cells were more efficient in restraining tumours in an AT3-OVA mammary carcinoma model, expressing lower frequencies of PD-1 or LAG-3 indicating that they may be less susceptible to exhaustion [161]. A novel small molecule inhibitor of PTPN22, named L-1 has promoted anti-tumour immune responses dependent upon the activation of CD8^+^ T-cells and macrophage polarisation towards the anti-tumour M1 phenotype [162]. In humans, an SNP in PTPN22, R620W, is reduced in nonmelanoma skin cancer patients, whilst its presence is associated with ICB efficacy [162]. Patients with a variant of PTPN22, rs2476601, are typically observed less frequently compared to that of healthy controls and respond significantly better to checkpoint inhibitor immunotherapy [163], observations supporting the potential of targeting PTPN22.

The antigen-specific activation of CD8^+^ T-cells requires the presence of cholesterol to support TCR clustering, signalling and efficient formation of the immunological synapse. However, T-cell activation results in the esterification of cholesterol by Ac-CoA acetyltransferase (ACAT)-1, reducing subsequent activation. Pharmacological inhibition of ACAT-1 by Avasimibe led to potentiated effector function and enhanced proliferation of CD8^+^ T-cells due to the increases in the levels of cholesterol in plasma membranes. The use of Avasimibe in a murine model of melanoma induced anti-tumour immunity which was potentiated in combination with anti-PD-1 ICB [164]. A peptide vaccine targeting Kras, when combined with Avasimibe, facilitated CD8^+^ T-cell tumour infiltration, IFN-γ expression and decreased the presence of regulatory T-cells in the TME and improved control of tumours in a lung cancer mouse model [165].

SHP-1 is progressively increased in activated T-cells and inhibition of SHP-1 using the PTP inhibitor sodium stibogluconate results in increased T-cell degranulation and cytotoxicity [166]. In a murine melanoma model expressing OVA antigen, anti-PD-1 therapy could enhance responses by high affinity T-cells however SHP-1 knockdown of OT-I CD8^+^ T-cells, combined with anti-PD-1 ICB, mediated long-lasting suppression of tumour growth which was mediated by low-affinity T-cells. This indicates that targeting SHP-1, like increased response to weak antigen in PTNT22 knockouts, could broaden the T-cell repertoire capable of targeting tumours [167]. These findings are important since, while ICB effectively slows tumour progression and increases T-cell frequencies, a study of murine tumour-infiltrating T-cells showed that the diversity of intratumoral T-cells remained stable, indicating that ICB does not broaden the T-cell repertoire within murine melanoma tumours, but rather expands existing T-cell populations and enhances their effector capabilities [168]. It is unclear whether inhibition of PTPN22 or SHP1 alone can significantly improve anti-tumour CD8^+^ T-cell responses however in combination with ICB in murine models has demonstrated improved efficacy compared to monotherapy. Systemically targeting tyrosine phosphatases such as PTPN22 or SHP-1 is complicated by the broad expression of these proteins and thus the possibility of increasing the function T-regs, potential off target effects and autoimmune toxicities. Therefore, such strategies may be particularly suited to immunotherapy based upon adoptive cell transfer or the development of mechanisms to specifically target CD8^+^ T-cells [169].

**Mitogen activated protein kinase.** The MAPK pathways involve RAS/RAF, Phosphoinositide 3-kinases (PI3K), Mitogen-activated protein kinase (MEK) and extracellular-signal-regulated kinase (ERK) signalling. This pathway is overactivated in numerous tumours, such as those with B-RAF mutations, and presents a therapeutic target. B-RAF and MEK inhibitors block the MAPK signalling pathway, induce apoptosis and have potential in clinical use for cancer treatment [170]. However, MAPK signalling also acts downstream of the TCR complex and is essential for T-cell activation, proliferation, survival and differentiation and MAPK inhibition can inhibit T-cell proliferation and cytokine production.

Use of BRAF inhibitors was shown to enhance the immunogenicity of Melanoma without negatively effecting T-cell function [171,172] possibly through increases in HLA expression [173], however, other studies observed reduced TIL in BRAF treated tumour models [174]. MEK inhibition with Trametinib was shown to upregulate tumour surface expression of MHC and PD-L1 in TNBC cells, resulting in increased TIL in a murine model of breast cancer. Combining MEK inhibition with PD-1 ICB demonstrated enhanced anti-tumour immune responses [175]. In another study MEK inhibition transiently inhibited T-cell functions but was synergistic with anti-CTLA-4, anti-PD-1 or anti-PD-L1 blockade [176]. Single-agent BRAF inhibition increased the presence of TAMs and Tregs in tumours. This could be reduced with duel MEK inhibition which, alongside anti-PD-1 ICB immunotherapy, promoted T-cell infiltration, intertumoural IFN-γ, improved cytotoxicity and tumour regression [177]. In contrast, a study using a murine model of TNBC showed that MEK inhibition adversely effects of TIL frequency, proliferation, and cytokine expression, likely due to the role of MEK in downstream TCR signalling. The use of anti-4-1BB/anti-OX-40 T-cell agonist antibodies rescued effects of MEK inhibition on T-cells via redirection of MAPK signalling towards increased downstream activation of p38/JNK and PI3K/AKT pathways, resulting in augmented anti-tumour effects which significantly prolonged survival in immunocompetent mouse models of TNBC [178]. The combination with anti-PD-1 ICB plus agonist immunotherapy with MEK inhibition in these models demonstrated significantly improved survival outcomes, compared to that of MEK-inhibition or agonist immunotherapy alone. In these experiments the MEK inhibition-dependent upregulation of MHC on tumours was further enhanced by immunotherapy dependent expression of IFN-γ, providing a mechanism by which combined therapy synergised to enhance T-cell responses. This is an example of plasticity within TCR signalling through the use of T-cell agonists which allows for tumour cell specific inhibition without inhibiting anti-tumour immunity and which mirrors the plasticity in glutamine blockade discussed below. Interestingly the addition of IL-15 can also overcome the immunosuppressive effect of MEK inhibition on T-cell activity [179]. Another study showed that MEK inhibition blocked naïve CD8^+^ T-cell priming in tumour-bearing mice whilst increasing the number of antigen specific effector CD8^+^ T-cells within the tumour. In this study MEK inhibition protected CD8^+^ TIL from cell death driven by chronic TCR stimulation and maintained cytotoxicity. The combination of MEK inhibition and PDL-1 ICB induced synergistic and durable regression of tumours compared to the modest effects of single agent therapy [180]. This study indicates that MEK inhibition can have a net positive effect on anti-tumour immune responses however timing is important to avoid blocking the generation of effector T-cells. These observations are supported by a study using short term treatment with combined BRAF and MEK inhibitors in a murine model of BRAF mutant melanoma. Short term treatment enhanced tumour infiltration of CD8^+^ T-cells, which subsequently decreased, but combining short term BRAF and MEK inhibitor therapy with anti-PD-1 ICB increased their anti-tumour activity [181]. In humans, 15 patients with BRAF V600-mutated metastatic melanoma were studied in a clinical trial involving the BRAF inhibitor dabrafenib, trametinib, and pembrolizumab as triple-combination therapy which may benefit this subset of patients by increasing the frequency of long-lasting anti-tumour responses. Increases in MHC class I expression and immune infiltration induced by triple therapy support observations that MAPK inhibition does not negatively impact the generation of an intratumoral immune response [182]. MEK inhibition also reduces the frequency of Tregs, possibly through reductions in the expression of IL-10 and TGF-β from tumour cells in addition to TAMS and MDSC, potentially due to these subsets being driven by MAPK signalling [178].

Numerous ERK-inhibitors are in clinical development for their ability to suppress tumour progression, but they may also affect T-cell responses. ERK acts downstream of BRAF and MEK in TCR signalling to direct the activation and cytokine expression of T-cells [183] and is particularly important in maintaining the functionality of effector CD8^+^ T-cells [184]. The triggering of the TCR signalling pathway results in extensive changes in protein expression, many of which are independent of ERK signalling, yet significant positive and negative changes in protein expression are regulated via ERK, including transcription factors and cytokines important in mediating T-cell immune responses and communication [185]. ERK-inhibitors are yet to be studied in the context of T-cell immunotherapy although a trial involving an ERK inhibitor and the Anti-PD-1 ICB pembrolizumab is underway (NCT02902042). A study of the BRAF inhibitor Dabrafenib showed that it can enhance pERK expression levels and did not suppress human CD4^+^ or CD8^+^ T-cell function. In contrast the MEK inhibitor Trametinib reduced pERK levels, and resulted in partial inhibition of T-cell proliferation and expression of a cytokine and immunomodulatory gene subset. Interestingly Trametinib effects were partially offset by adding dabrafenib [176].

PI3K signaling has emerged as an important factor in the control of CD8^+^ T-cell signaling. The PI3K delta subunit coordinates transcriptional, chromatin, and metabolic changes characterised by enhanced mTORC1 and c-Myc signatures to promote effector CD8^+^ T cells. This is associated increases in FAS-mediated apoptosis, reductions in transcription factors including TCF-1 and reductions in the generation of central memory [186]. The role of the PI3K delta subunit appears to have profound, differential effects CD8^+^ T-cells, controlling their fate, and represents a novel target for cancer immunotherapy.

P38 regulates T-cell activation via selective activation of NFATc and is responsible for regulating T-reg function [187]. Activation of p38 mitogen-activated protein kinase is critical step for acquisition of effector function in cytokine-activated T-cells, but it acts as a negative regulator in T-cells activated through the T-cell receptor and suppresses T-cell proliferation [188]. P38 negatively regulated the ERK-dependent IL-2 transcription in T-cells stimulated by anti-CD3/anti-CD28 stimulation [189]. In a study using a CRISPR-Cas9-based genetic screen of primary T-cells the impact of disrupting 25 T-cell receptor-driven kinases was measured and p38 was identified in effecting T-cell activation and differentiation. Pharmacological inhibition of p38 using BIRB796 increased cell expansion and memory while reducing oxidative and genomic stress, improving the efficacy of murine anti-tumour T-cell responses [190] consistent with an earlier study showing that blocking p38 MAPK signalling in T-cells with senescent phenotypes enhanced proliferation and reduced TNF-α expression [191]. Another study using the p38 MAPK Inhibitor SB203580 demonstrated inhibition of TNF-α mediated expansion of Mouse CD4+Foxp3+ Regulatory T-cells [192]. Whilst p38 inhibition may be a promising strategy to support T-cell function P38 may act as a tumour suppressor under particular circumstances and more study is needed to determine how best to regulate its function [193].

Recent studies targeted other components of TCR-MAPK signalling. Diacylglycerol kinase (DGK) α limits the extent of Ras activation in response to antigen recognition and its upregulation facilitates a hypofunctional, exhausted T-cell state. Pharmacological DGKα targeting restores cytotoxic function of CAR T-cells and TIL isolated from solid tumours, suggesting a mechanism to reverse T-cell exhausted phenotypes. Pharmacological DGKα inhibition selectively enhances AP-1 transcription and combines with anti-PD-1 ICB [194]. In another recent study, inhibition of Src Homology Region 2-Containing Protein Tyrosine Phosphatase (SHP)-2 within NSCLC tumours using SHP099 results in increased TIL but also increased intratumoural MDSC via increased production of CXCR2 ligands which could be prevented through the use of combined CXCR2 inhibition [195].

**Mammalian target of rapamycin (mTOR):** The protein kinase mTOR is a central component of cell growth regulation which integrates environmental cues to coordinate diverse cellular processes [196]. T-cell activation switches on the PI3K/ATK/mTOR signalling pathway which promotes T-cell effector differentiation by rewiring metabolism to enable growth, protein translation and function in proliferating cells. In T-cells mTOR is an important metabolic sensor forming two mTOR complexes, mTORC1 and 2. These complexes are distinguished by their accessory proteins, differential sensitivity to rapamycin, and their unique substrates and functions. MTORC1 senses environmental conditions to stimulate cell growth and effector function through protein synthesis and inhibition of autophagy. For example, the withdrawal of glucose, amino acids or oxygen leads to rapid suppression of mTORC1 activity [197]. In contrast mTORC2 is activated by growth factors to promote survival and proliferation. They have separate and perhaps opposing roles in modulating CD4^+^ T-cell differentiation into Th1, Th2, Th17, or T-reg subsets whilst modulating the differentiation of CD8^+^ T-cells from effector to memory subsets. Although mTOR inhibitors have originally been used as immune suppressors through the generation of T-regs and reductions in T-cell proliferation, the inhibition of mTOR has both immune suppressive and activatory effects and, under particular conditions and depending upon whether mTORC1 or mTORC2 are inhibited, mTOR inhibition can enhance anti-tumour immune responses [198]. Thus, methods to modulate this pathway are under investigation to calibrate T-cell differentiation and promote the generation of lasting memory T-cells.

Methods to inhibit mTOR include the inhibition of PI3K [186,199], which generates T-cells with a less differentiated state, or mTOR inhibition using rapamycin or its analogs (rapalogs), which enhance T-cell survival, increase mitochondrial respiration and support the generation of persistent memory T-cells [200]. This is hypothesized to be due to enhanced autophagy generating improved mitochondrial fitness [201,202]. In a preclinical study, the rapalog Temsirolimus increased the anticancer efficacy of heat-shock protein-based vaccines in established renal cell carcinoma and melanoma in mice [203]. In another study of mice bearing tumours expressing ovalbumin antigen (OVA), vaccination with an OVA-based vaccine demonstrated significantly improved reductions in tumour growth with the subsequent addition of Temsirolimus compared to that of the vaccine or Temsirolimus-only groups [204]. This effect was associated with a higher percentage of OVA-specific CD8^+^ T-cells in the tumour microenvironment with a phenotype of Tcm-cells. These data provide support for the use of rapalogs to improve cancer vaccination. However, T-regs generated through mTOR inhibition can act to inhibit the anti-tumour immunity [205] and the addition of rapamycin to cancer vaccines also demonstrated an inhibitory effect [206] emphasising that the immune modulating properties of mTOR inhibitors have to be carefully considered in the design of immunotherapies. This is supported by a study testing the effects of rapamycin on both the priming/expansion and contraction phases of anti-tumour vaccination. Rapamycin induced a marked contraction of antigen specific CD8^+^ T-cells when used during priming but an increased number of antigen-specific memory CD8^+^ T-cells and improved efficacy when used in the contraction phase [207].

In human studies, a recent report investigating the use of the rapalog sirolimus found that it inhibited transplant rejection whilst promoting prolonged efficacy of anti-PD-1 ICB in transplant patients treated for cancer [208]. In support of the dual action of mTOR inhibition the rapalog everolimus could simultaneously promote expansion of T-regs and activate tumour-specific Th1 immunity [209]. Inhibition of mTOR can result in lower effector CD8^+^ T-cell responses but improved differentiation of memory cells [198], so it is important to balance the use of rapalogs to activate rather than suppress immune responses. Conjugation of a raptor targeting siRNA to an aptamer that binds 4-1BB specifically inhibits mTORC1 activity in activated CD8^+^ T-cells. This 4-1BB aptamer-raptor siRNA conjugate was superior to systemic rapamycin in terms of protective anti-tumour immunity, whilst its specificity for activated 4-1BBhi CD8^+^ supports the formation of memory cells whilst leaving 4-1BB^lo^ T-cells to induce effector responses [210].

Vitamins and their metabolites have fundamental effects on T-cell homeostasis. Vitamin D modulates both inflammation and immunity and disruption in vitamin D metabolism has the potential to alter anti-tumour immune responses [211]. T-cells convert inactive 25(OH)D3 into its active metabolite 1,25(OH)2D3, using the enzyme CYP27B1, allowing 1,25(OH)2D3 to bind to the vitamin D receptor (VDR) a transcriptional regulator expressed in most leukocytes. The VDR can subsequently bind to genes with vitamin D responsive elements to regulate gene expression [212]. This results in the upregulation of numerous genes including the enzyme phosphoinositide phospholipase C-γ1 (PLC-γ1), a component of the TCR signalling pathway in T-cells, thus promoting increased TCR activation. TCR signalling can in turn induce the expression of the VDR [212,213] in a positive feedback loop. The VDR can subsequently regulate the expression of IL-2 and T-cell proliferation [214], acting as a break on T-cell responses. Vitamin D signalling can also regulate the expression of its own receptor and the expression of CYP27B1 [215] to calibrate levels of 1,25(OH)2D3 and VDR activity. The VDR can subsequently also induce the expression of the 1,25(OH)2D3 degrading enzyme CYP24A1 [216]. This negative feedback via degradation of 1,25(OH)2D3 can in turn be mitigated by IFN-γ signalling to increase the expression of the VDR [217]. Thus 1,25(OH)2D3 acts as a self-correcting calibrator of T-cell function capable of modulating the activation of T-cells whilst regulating both its receptor and its abundance and activation within cells. These mechanisms identify vitamin D as a master regulator of T-cell homeostasis, a theory supported by clinical observations of perturbations in the regulation of vitamin D. Cancer patients and immune suppressed individuals were found to have a high prevalence of vitamin D deficiency [218]. In contrast low serum levels of vitamin D are associated with the development of autoimmune disease [219,220] or increased susceptibility to infectious disease such as respiratory tract infections [221,222,223]. VDR knockout mice show increased sensitivity to autoimmune diseases and are more prone to experimentally induced tumours [224]. Vitamin D-binding protein (VDBP) sequesters vitamin D from T-cells and reduced serum VDBP is strongly associated with longer overall survival in advanced renal cancer patients treated with anti-PD-L1 ICB [225] consistent with an association between vitamin D deficiency, worse outcomes in patients with stage IV metastatic melanoma [226] and observations that vitamin D intake (around 1000 IU) before ICB therapy in melanoma patients reduced the risk of developing colitis by altering the Th1/Th2 ratio [227].

Observations that 1,25(OH)2D3 or VDR deficiency is associated with increased susceptibility to tumours and autoimmune disease, both due to the action of Th1 based cellular immunity, suggests that 1,25(OH)2D3 acts to reset T-cell responses to an immunological baseline, limiting hyper-activated T-cell responses in autoimmunity whilst preventing the exhaustion or skewing of anti-tumour T-cell responses in the setting of cancer.

All trans Retinoic acid (ATRA), a metabolite of vitamin A, is the major active immune modulating metabolite derived from Carotenoids [228]. Like 1,25(OH)2D3, ATRA is exhibits both activatory and inhibitory effects on T-cell function. ATRA production by DC can inhibit the differentiation of naïve T-cells towards a Th17 phenotype by blocking IL-6, IL-21 and IL-23 expression. Stimulation of DC by ATRA can in turn induce DC dependent IL-10 and TGF-β expression from T-regs [229]. Yet in the setting of inflammation ATRA may activate Th1 based immune responses or, in the presence of IL-4, enhance Th2 induction. ATRA, through its nuclear receptor retinoic acid receptor alpha (RARα), also sustains stable expression of Th1 transcriptional programs whilst repressing genes associated with Th17 differentiation. As such, ATRA signalling is essential for limiting Th1-cell conversion into Th17 effectors and for preventing pathogenic Th17 responses in vivo [230] and blockade of RARα results in defects in proinflammatory Th1 activation via impaired calcium mobilization [231]. Notably, ATRA may have a dose dependent effect, inhibiting Th17 cells in favour of induction of T-regs at higher doses (>10 nM) but favouring the generation of Th17 cells at lower doses or particular inflammatory environments [232], reminiscent of the role of 1,25(OH)2D3 differentially regulating T-cell activation based upon cellular concentration. In fact, ATRA induces the expression of CYP26b1, which degrades ATRA, whilst TGF-β inhibits it and prolongs the accumulation of ATRA [233], a mechanism similar to that of the differential regulation of 1,25(OH)2D3 by IFN-γ and CYP24A1. Each vitamin metabolite may also have biphasic effects, priming Th1-based immune responses at baseline and subsequently regulating their magnitude [231]. This ability to calibrate T-cell immunity may limit tissue damage and prevent the onset of exhaustion.

In cancer, colitis induced dysbiosis was shown to induce deficiency in colonic ATRA resulting in the promotion of tumorigenesis whilst supplementation of ATRA resulted in CD8^+^ T-cell dependent reductions in tumour burden through the upregulation of HLA on tumour cells [234]. Encapsulated ATRA in pegylated liposomes was shown to accumulate inside tumours resulting in increased delivery and duration of ATRA capable of inhibiting MDSC and synergizing with anti-PD-1 treatment to result in increased CD8 T-cell infiltration [235]. In support of this are observations that exogenously administered ATRA greatly increased both the numbers and cytotoxic functions of effector and memory CD8 T cells in intestinal mucosal tissues, in a model of LCMV vaccination [236]. In contrast, murine models of sarcoma have shown that ATRA derived from tumours stimulated by T-cell dependent expression of IL-13, which enhances immune suppression within the TME by polarising monocytes toward suppressive TAM cells rather than APC [237]. Collectively these studies indicate the ATRA may act as an amplifier of immune cell differentiation dependent upon the environment in which it acts, whilst being capable of exerting negative feedback as ATRA concentration increases. ATRA binding of RARα allows it to interact with transcription factors including AP-1, whose function is inhibited and N-FAT isoforms whose expression is regulated. In each case the interaction is reciprocal and an example of the complexity of ATRA signalling, indicating that its effects on T-cell function and potential as an immunotherapy are as yet incompletely understood [238].

**Table 4 vaccines-09-01392-t004:** T-cell signalling targets under investigation as cancer immunotherapies.

T-Cell Signalling	Effect on T-Cells	Questions	Ref
PTPN22 inhibition	small molecule inhibitor of PTPN22, named L-1 has promoted anti-tumour immune responses dependent upon the activation of CD8^+^ T-cells	Not yet studied in clinical trials	[162]
Cholesterol metabolism	Increasing cholesterol availability by blocking ACAT-1 with avasimibe leads to potentiated effector function and enhanced proliferation of CD8^+^ T-cells	Not yet studied in clinical trials	[164,165]
SHP-1 inhibition	inhibition of SHP-1 using the PTP inhibitor sodium stibogluconate results in increased T-cell degranulation and cytotoxicity	Increasing the function T-regs, potential off target effects and autoimmune toxicities	[166,167]
SHP-2	Inhibition of Src Homology Region 2-Containing Protein Tyrosine Phosphatase (SHP)-2 within NSCLC tumours using SHP099 results in increased TIL	Potential to increase intratumoural MDSC via increased production of CXCR2 ligands	[195]
Diacylglycerol kinase-RAS	Pharmacological DGKα targeting restores cytotoxic function of chimeric antigen receptor and CD8^+^ T-cells isolated from solid tumours, suggesting a mechanism to reverse T-cell exhausted phenotypes.	Careful use of DGKα blockade will be required to prevent the inhibition of effector T-cell responses via MAPK pathway inhibition	[194]
MEK inhibition	Trametinib upregulate tumour surface expression of MHC and PD-L1 in TNBC cells, resulting in increased TIL in a murine model of breast cancer. Combining MEK inhibition with PD-L1/PD-1 ICB demonstrated enhanced anti-tumour immune responses	MEK inhibition adversely effects of TIL frequency, proliferation and cytokine expression	[175,178,180]
P38 inhibition	Pharmacological inhibition of p38 using BIRB796 increased cell expansion and memory while reducing oxidative and genomic stress, improving the efficacy of murine anti-tumour T-cells	Effect on tumour cells is not clear and inhibition may have a protumorigenic effect.	[189,190,191,192,193]
Rapalogs	Can increase memory CD8^+^ T-cells and enhance anti-tumour T-cell based immune responses	May reduce the function of effector T-cells and promote expansion of regulatory CD4+ T-cells. The sequence of mTOR inhibition relative to vaccination or use of ICB is yet to be fully understood.	[197,198,199,200,201,202,203,204,205,206,207,208,209,210]
Vitamin D	Regulates T-cell activation, proliferation and cytokine expression	How effective is the therapeutic use of vitamin D?	[212,225,226,227]
All trans Retinoic acid	Pleiotropic modulation of Cd4+ T-cell differentiation and priming effect on CD8^+^ T-cells	Delivery of ATRA (systemic or intratumourol?) and potential as a combination with ICB need further study	[228,229,230,231,232,233,234,235,236,237,238]

## 6. Mitochondrial Function

T-cell metabolism is essential to the generation and maintenance of effective T-cell responses. Naïve T-cells utilise oxidative phosphorylation and, upon recognition by the TCR of a cognate peptide/MHC complex, undergoes a metabolic switch to utilises glycolysis and differentiate into effector T-cells. The enzymes used in the glycolysis pathway are necessary to induce the effector functions of activated T-cells, mechanistically linking energy production with T-cell function [239]. Mitochondrial remodelling is required to adapt to the specialized metabolic needs for effector and memory T-cells. For effector cells mitochondrial fragmentation produces punctate mitochondria with loose cristae exhibiting inefficient electron transport but a high capacity to buffer calcium, resulting in increased anaerobic glycolysis and activation of NFAT required for T-cell activation. In contrast memory T-cells have fused mitochondria and tight cristae organization which promotes efficient electron transport activity and favourable redox balance, allowing for the continued transfer of pyruvate into mitochondria for OXPHOS and FAO. The importance of mitochondrial remodelling on T-cell function is demonstrated by the forced fusion of mitochondria in effector T-cells using the small molecule fusion promoter M1 [208]. Mitochondrial fusion in effector T-cells generated an increased oxidative capacity in mitochondria which supported T-cell persistence, while higher aerobic glycolysis supported increased cytokine production, which were thought to result in superior anti-tumour function. Thus, mitochondrial remodelling acts as a signalling mechanism directing T-cell metabolic programming and effector function [240]. In the TME T-cells can exhibit dysfunctional mitochondria, which inhibited their anti-tumour function and led to T-cell exhaustion. These TIL accumulate depolarized mitochondria, due a decrease in mitophagy of dysfunctional mitochondria, resulting in functional, transcriptomic, and epigenetic changes indicative of terminally exhausted T-cells and caused by a combination of metabolic stresses alongside TCR and PD-1 signalling. Mitochondrial dysfunction can result in these T-cells becoming locked in a permanent dysfunctional state [202]. Evidence that altering T-cell metabolism can reinvigorate such dysfunctional T-cell responses comes from observations that Chronic AKT signalling in TIL induces progressive loss of PPAR-gamma coactivator 1α (PGC1α), responsible for programming mitochondrial biogenesis, but enforced expression of PGC1α induces improved metabolic and T-cell effector function [241]. The central role of mitochondria to T-cell function inspired efforts to therapeutically regulate mitochondrial function and cellular metabolism to improve cancer immunotherapy.

**5′ adenosine monophosphate-activated protein kinase (AMPK) signalling.** AMPK is a nutrient and energy sensor that functions to maintain energy homeostasis [242]. AMPK is activated under conditions of low intracellular ATP following stresses such as nutrient deprivation or hypoxia. AMPK controls cellular growth through inhibition of mTOR [243], specifically inhibiting mTORC1 signalling in part by phosphorylating mTORC1 subunit Raptor [244] and promoting OXPHOS. AMPK can induce mitochondrial remodelling via the activation of Sirtuin (SIRT)-3 leading to Optic Atrophy (Opa)-1 dependent mitochondrial remodelling [245]. AMPK also activates peroxisome proliferator-activated coactivator 1α (PGC1α), the master regulator of mitochondrial biogenesis and function, by regulating the intracellular levels of NAD+, required as a substrate for the deacetylase Sirt1, which is involved in mitochondrial biogenesis, switching on oxidative phosphorylation [246], vital to the generation of effective anti-tumour immunity in the TME [241].

Metformin is an antidiabetic drug which has demonstrated T-cell modulating properties via activating AMPK. In murine models, Metformin increased the number of infiltrating CD8^+^ Tem-Cells and IL-10+ T-regs in tumour transplanted mice. It also decreased the levels of MDSC and Th17 T-cells and induced an anti-tumour immune response [247]. In a murine cancer vaccine study, administration of metformin after vaccination showed an increase in CD8^+^ memory T-cells, dependent upon increased mitochondrial FAO, which conferred protective immunity upon subsequent tumour challenge. In this model metformin was administered after the peak of the effector T-cell response and prior to inoculation with tumour cells. Six out of nine metformin-treated mice survived >33 days compared to only 1/8 PBS-treated mice which correlated with an increase in memory T-cells [248]. In human’s frequencies of Tscm and Tcm-cells increased both in peripheral and tumour-infiltrating CD8^+^ T-cell populations in metformin-treated lung cancer patients compared with those not taking metformin. In vitro experiments demonstrate that metformin can promote the formation of memory CD8^+^ T-cells with enhanced survival and reduced the expression of PD-1 through increased expression of the transcription factor EOMES [249]. Metformin is being studied in combination with the anti-PD-1 ICB with Pembrolizumab, platinum-based immunotherapy and fasting-mimicking diet in immune suppressive, metabolically vulnerable LKB1-inactive lung adenocarcinoma. This trial is one of the first its kind and will offer evidence on the efficacy of immunotherapy whilst applying metabolic stress and also of the ability of metformin to simultaneously enhance chemotherapy and immunotherapy (NCT03709147).

**Metabolites:** Nicotinamide adenine dinucleotide (NAD+), a metabolite of niacin (Vitamin B3) modulates T-cell activation and differentiation by regulating mitochondrial energy production. In CAR-T-cells increased NAD+ levels via overexpression of nicotinamide phosphoribosyltransferase could enhance tumour cell cytotoxicity and elevate cytokine production. Treatment of tumour bearing mice with nicotinamide (NAM), an amide derivative of niacin, resulted in increases in intracellular NAD+ concentrations. The supplementation of mice with NAM alone did not affect tumour growth however CAR-T plus NAM supplementation demonstrated improved CAR T-cell responses capable of eliminating tumours. NAM supplementation also significantly extended the survival time of mice, associated with increased TIL in tumours of treated mice. NAM supplementation was also capable of improving anti-PD-1-based immunotherapy in tumours otherwise refractory to anti-PD-1 treatment, effectively inhibiting tumour growth and significantly extending the survival time of the mice, again associated with significantly increased presence of TILs [250]. NAM treatment can attenuate the increase of mitochondrial content and reactive oxygen species (ROS) in T-cells activated by CD3/CD28 antibodies. This is accompanied by accelerated clonal expansion due to attenuated apoptosis involving reduced ROS-triggered proapoptotic events and upregulation of Bcl-2 expression albeit with lower cytokine expression [251]. The accumulation of depolarised mitochondria in anti-tumour T-cells contributes to exhaustion whilst supplementation with nicotinamide riboside reduced the accumulation of depolarized mitochondria in T-cells via Drp-1-dependent mitophagy, improving T-cell mitochondrial fitness and responses to anti-PD-1 ICB in OVA melanoma-engrafted mice by [202].

L-arginine is also utilised during T-cell metabolism and its supplementation improves anti-tumour T-cell responses by enhancing memory formation and mitochondrial respiration. Activation of T-cells leads to increases in arginine metabolism precipitating a drop in intracellular L-arginine concentration. L-arginine supplementation results in metabolic changes including a shift from glycolysis to oxidative phosphorylation, counteracting the Warburg effect, possibly due to increases in the serine biosynthesis pathway. This promotes the generation T-cells with a central memory phenotype, increased expression of CCR7 and CD62L, reduced IFN-γ expression and greater persistence and survival, possibly mediated by the transcriptional regulators BAZ1B, PSIP1, and TSN, capable of sensing L-arginine levels [252]. In a murine tumour model these T-cells demonstrated anti-tumour responses. In support of these observations L-arginine, in the setting of bronchial inflammation, can improve mitochondrial function in lung epithelial cells. Reduced L-arginine bioavailability leads to increased formation of peroxynitrite, which could induce mitochondrial dysfunction, whereas L-arginine supplementation increased mitochondrial cytochrome c oxidase activity and ATP levels [253].

Tumour cells require glucose and glutamine for anabolic metabolism, growth, and proliferation. These processes also create the hypoxic, acidic, nutrient-depleted TME, which acts to inhibit responses by infiltrating T-cells. Glutamine blockade in tumour baring mice using JHU083 which is capable of blocking a broad range of glutamine-requiring enzymes, results in blockade of glutamine metabolism, inhibiting glucose metabolism through the TCA cycle, glycolysis and related pathways, thus disabling Warburg physiology. This effect inhibits tumour cell viability, proliferation, and cell cycle progression whilst suppressing oxidative and glycolytic metabolism of cancer cells, leading to decreased hypoxia, acidosis, and nutrient depletion. In contrast T-cells respond to glutamine blockade by up-regulating oxidative metabolism and adopting a long-lived, highly activated phenotype indicating that glutamine blockade has both anti-tumour and T-cell activatory properties leading it to be described as a ‘metabolic checkpoint for cancer immunotherapy [254].

The metabolism of amino acids such as arginine, glutamine, serine, methionine, tryptophan and leucine are vital to the survival and proliferation of cancers [255]. Catabolism of amino acids such as arginine and tryptophan also act to inhibit anti-tumour T-cell function whilst T-cells require many of the same amino acids for activation and differentiation. An understanding of the amino acid metabolic needs of specific tumours, and catabolism by tumour cells or suppressive immune subsets such as MDSC, will be vital to design strategies to inhibit these pathways through amino acid depletion or blockade whilst maintaining the availability of amino acids for T-cell function [256].

Inosine, which can be produced by the gut microbiome, is an alternative substrate for T-cell growth and function. T-cells metabolize inosine into hypoxanthine and phosphorylated ribose by purine nucleoside phosphorylase. The ribose subunit of inosine can then be utilised in metabolic pathways to provide ATP and biosynthetic precursors. Some tumour cells are capable of utilizing inosine as a carbon source but the supplementation with inosine was shown to enhance the anti-tumour efficacy of ICB and adoptive T-cell transfer in solid tumours that are defective in metabolizing inosine, providing evidence that, in tumours incapable of metabolising inosine, it may be useful to relieve tumour-imposed metabolic restrictions on T-cells [257].

Taken together, these studies demonstrate that modulating the metabolic state of T-cells has the potential to prime them for sustained anti-tumour effector responses, or promote the generation of memory. However, it will be necessary to strike a balance between effector and memory T-cell differantiation and to avoid providing tumour cells with pro survival metabolic cues. Some of the promising approaches described here are summarised in Table 5.

## 7. Summary

The ability of T-cells to respond to antigenic challenge, referred to as the ‘immune set point’ is calibrated by numerous factors which are introduced by our diets, the gut microbiome, underlying inflammation, age associated immune-senescence, innate immune activation, or inhibition within the TME. Many of these factors are being studied either as targets or agents for combination therapies, particularly involving vaccination, ICB, or adoptive T-cell therapy. Knowledge of the immune processes involved in the activation and subsequent differentiation of effector T-cells to either long lasting memory or exhaustion, and the role of mitochondrial biogenesis and metabolism in these processes, should provide scope for more elaborate combination therapies which incorporate markers of pretherapy immune function and consider tumour antigenicity, T-cell priming, and the maintenance of memory T-cells into the schedule of future combinations (Figure 3). Optimal anti-tumour immunity must balance effector function and the generation of immune memory. Excessive effector function may reduce the generation of sustained memory T-cell immunity. This may be through the generation of exhaustion or the failure of mechanisms intended to drive memory responses such as an inability to control of effector T-cell activation or the disruption in glycolysis/OXPHOS due, for example, through inadequate availability of L-arginine, which may promote irreversible dysfunction of anti-tumour CTL. Thus, the rational design of cancer vaccination or immunotherapy will need to consider both priming and maintenance phases. These may be served by the same agent, such as IL-21, or require different agents capable of advancing T-cell activation and anti-tumour effector functions based upon the utilisation of glycolysis before inducing a metabolic switch to promote OXPHOS, FAO and the differentiation into long lived memory T-cells. This approach is made possible by studies performed over the last twenty years that have characterised the association between cellular metabolism and T-cell effector function and differentiation and revealed the metabolic competition within the TME between tumour cells and T-cells exemplified by the study of AMPK, mTOR, glutamine blockade, L-arginine levels, supplementation with NAD+, and Methionine restriction.

Combination immunotherapy is beginning to be rationally designed into sequential, rather than concomitant, combinations for example with the use of cytokines or MEK inhibition with ICB therapy, or metformin, NSAID and mTOR inhibitors in relation to vaccination. Sequence will become vital to combinations involving three or more agents as demonstrated by the use of MEK inhibition, ICB immunotherapy, and OX40 agonist. This will be particularly relevant for therapeutic combinations targeting multiple cells types, for example, tumour cells, immune inhibitory cells, and CD8^+^ T-cells. For such approaches an understanding of how to attack tumours without suppressing anti-tumour immunity, or enhance T-cell activation without promoting tumorigenesis, will be vital to their success. An important avenue of future research will involve identifying where T-cells exhibit plasticity in their response whilst tumour cells demonstrate dependence, for example with glutamine blockade or the use of costimulation of T-cells to avoid the inhibitory effects of targeting MAPK pathways. A selection of clinical trials exemplifying the concepts detailed in this review are summarized in Table 6.

ICB immunotherapy failure may be either primary or acquired. Primary failure may be either intrinsic to the tumour, its microenvironment or intrinsic to the T-cell. Many of the agents described here do not have appreciable therapeutic efficacy as single agents but act to raise the immune set point by priming T-cells to respond. Butyrate and vitamin D are amongst the agents which support T-cell function. Butyrate and Vitamin D have complex roles in the regulation of immune responses, and it is notable that both agents are associated with improved efficacy of ICB despite their ability to inhibit T-cell proliferation, maintain immune inhibitory cells, such as T-regs and iNK cells, and dampen Th1-based immune responses, which may seem counter intuitive. However, the ability to inhibit, and in so doing reset or rebalance, existing T-cell activation may be an important prerequisite for efficacious vaccination and immunotherapy, particularly in the context of underlying inflammation and chronic antigenic stimulation which induces dysfunctional and exhausted immune responses before the additional immune stimulus of vaccination or immunotherapy is applied. Vitamin D can calibrate T-cell responses in part through regulating its own production whilst butyrate can set an immune equilibrium by reducing innate immune inflammatory reactions whilst inducing both CTL and IL-10 based T-cell responses [140,258,259].

Finally, a number of the agents detailed here such as vitamin D, Niacin, polyphenols, probiotics, prebiotics, dietary fibre, and other dietary approaches potentially capable of supporting immunotherapy and vaccination are typically administered orally and are widely available as supplements without prescription. These agents are characterised by little or no toxicity however, as powerful immunotherapy becomes a first-line treatment for more cancers, the ability of these agents to alter the course of immunotherapeutic responses, either enhancing or impairing their efficacy, is increased. A greater understanding of how they work will be required, both to improve responses to cancer immunotherapy and to prevent the inappropriate use of interventions that may hinder rather than help.

## Figures and Tables

**Figure 1 vaccines-09-01392-f001:**
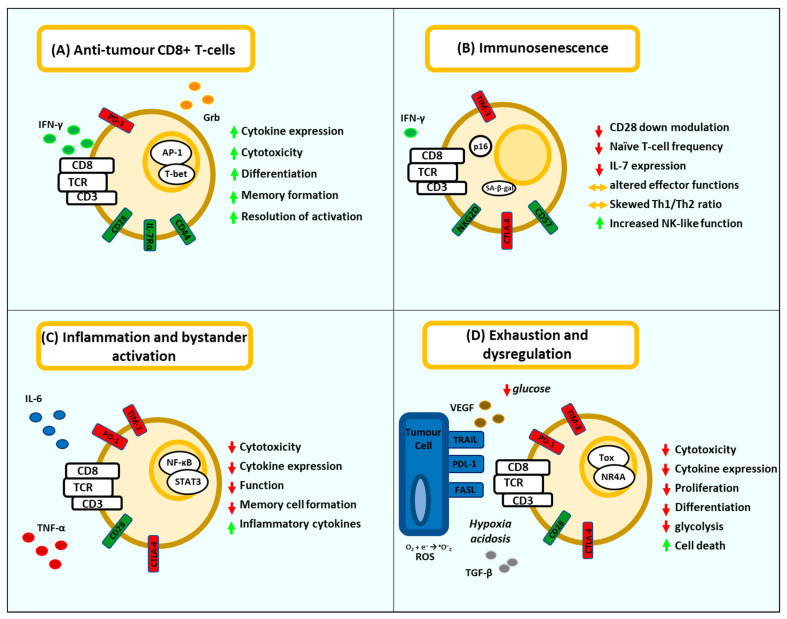
T-cell dysfunction. (**A**) Tumour-specific CD8^+^ T-cells need to acquire effector function upon ligation of their TCR with a cognate peptide-MHC complex. Effector function is characterised by expression of cytokines and cytotoxic granules including granzyme B and perforin. Effector T-cells must subsequently apoptosis or, upon appropriate stimulation which promotes metabolic changes, differentiate into long-lived memory cells upon resolution of antigenic challenge. (**B**) Immunosenescence is characterised by reduced TCR signal transduction, loss of CD28 expression, lower cytokine expression from CD8^+^ T-cells and a skewed Th1/Th2 ratio in CD4^+^ T-cells. Fewer naïve T-cells and increases in memory T-cells expressing NK-cell receptors and exhibiting increased innate-like effector functions are also characteristic of senescent T-cells. (**C**) Senescence is also associated with persistent inflammation. This inflammation also results from chronic infection leading to ‘bystander’ activation of T-cells by inflammatory cytokines. Signalling by cytokines such as IL-6 or TNF-α alter T-cell function, inhibit their differentiation into memory cells and induce apoptosis. (**D**) Dysregulation and exhaustion of TIL within TME shares some characteristics of senescence and bystander activation. TIL are subject to suppression via VEGF or TGF-β, suffer from metabolic dysregulation due to competition for glucose and amino acids and preventing differentiation into memory T-cells. TIL also express checkpoints due to chronic exposure to antigen. Ligation of these checkpoints with ligands expressed on tumour cells of suppressive immune cells, results in activation of programs of exhaustion based upon action of transcription factors Tox and NR4A blocking effector functions and potentially leading to anergy or apoptosis. Cytotoxic T-lymphocyte-associated protein 4 (CTLA-4), FAS ligand (FASL), Granzyme B (Grb), Interferon-γ (IFN-γ), Interleukin (IL), Nuclear factor kappa B (NF-κB), Programmed cell death protein 1 (PD-1), Programmed cell death protein ligand-1 (PDL-1), Reactive oxygen species (ROS), T-cell immunoglobulin and mucin-domain containing-3 (TIM-3), T-helper cell (Th), TNF-related apoptosis-inducing ligand (TRAIL), Transforming growth factor beta (TGF-β), Tumour necrosis factor-alpha (TNF-α), Vascular endothelial growth factor (VEGF).

**Figure 2 vaccines-09-01392-f002:**
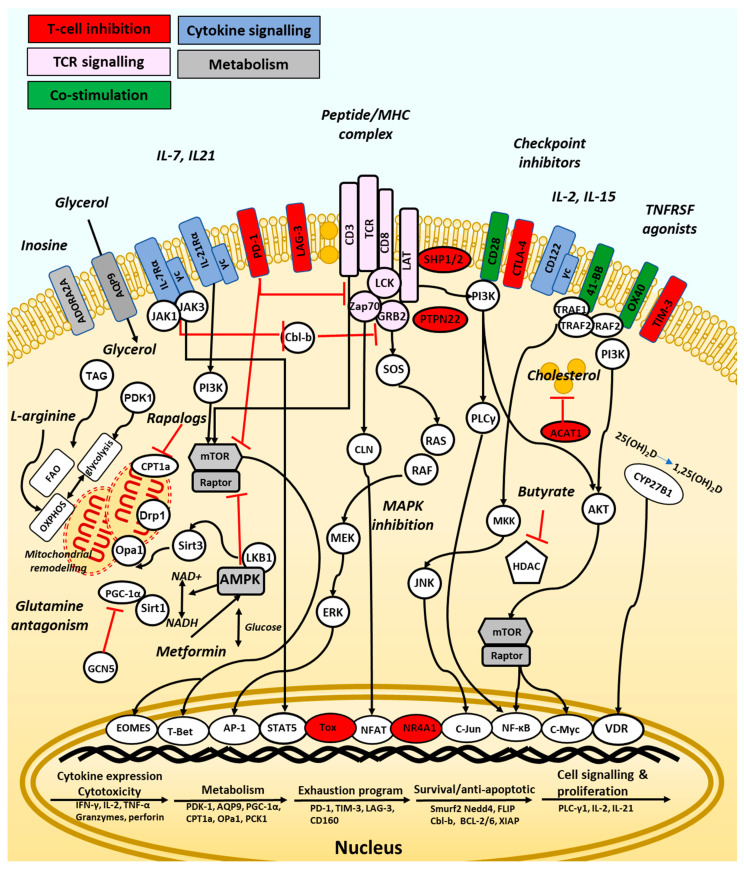
**Modulation of T-cell function for cancer immunotherapy**: T-cell modulation focuses on four aspects of T-cell biology. **TCR signalling** where inhibition of tyrosine phosphatases such as PTPN22 or SHP1/2 can potentiate T-cell activation and disruption of MAPK signalling may alleviate chronic antigenic stimulation and prevent exhaustion. **Co stimulation** either with common gamma chain cytokines or agonists of TNSFR receptor signalling may enhance TCR activation and effector function including activation of low affinity T-cells. Cytokines such as IL-7 or IL-21 may also promote proliferation, and subsequently, differentiation into long-lived memory T-cells. **Transcriptional regulation** of T-cell activation via HDAC-inhibition by butyrate or activation of VDR by 1,25(OH)2D (calcitriol) can regulate T-cell effector function and differentiation. Modulating **T-cell metabolism** through altering mitochondrial biogenesis to promote effector function or differentiation into memory cells through supplementation with L-arginine or NAD+, or through AMPK activation and mTOR inhibition. Activator protein 1 (AP-1), Adenosine A2a Receptor (ADORA2A), AMP-activated protein kinase (AMPK), Aquaporin-9 (AQP9), Calcineurin (CLN), Carnitine palmitoyltransferase (CPT)-1a, Cbl Proto-Oncogene B (Cbl-b), Cellular Myc (C-Myc), Cytotoxic T-lymphocyte-associated protein 4 (CTLA-4), Dynamin-related protein 1 (Drp1), Eomesodermin (EOMES), Extracellular-signal-regulated kinase (ERK), Fatty acid oxidation (FAO), General control non-depressible 5 (GCN5), Growth Factor Receptor Bound Protein 2 (GRB2), Histone deacetylase (HDAC), Janus associated kinase (JAK), Jun proto-oncogene (C-Jun), Lymphocyte activation gene 3 (LAG-3), Linker for activation of T cells (LAT) lymphocyte-specific protein tyrosine kinase (Lck), Liver kinase B1 (LKB1), Mammalian target of rapamycin (mTOR), Mitogen activated protein kinase (MAPK), Mitogen-activated protein kinase kinase (MEK), Nicotinamide adenine dinucleotide (NADH/NAD+), Nuclear factor kappa-light-chain-enhancer of activated B cells (NF-κB), Nuclear factor of activated T-cells (NFAT), Nuclear Receptor Subfamily 4 Group A Member 1 (NR4A1), Optic atrophy 1 (Opa-1), Oxidative phosphorylation (OXPHOS), Peroxisome proliferator-activated receptor-gamma coactivator-alpha (PGC-1α), phosphoenolpyruvate carboxykinase (PCK1), Phosphoinositide 3-kinases (PI3K), Phosphoinositide pho spholipase C-γ (PLCγ), Programmed cell death protein 1 (PD-1), Protein kinase B (AKT), Protein Tyrosine Phosphatase Non-Receptor Type 22 (PTPN22), Pyruvate Dehydrogenase Kinase 1 (PDK1), Rapidly Accelerated Fibrosarcoma proto-oncogene (RAF), Rat sarcoma virus (RAS), Signal transducer and activator of transcription-5 (STAT5), Sirtuin-1 (Sirt1), Son of sevenless (SOS), Src homology region 2 domain-containing phosphatase (SHP), T cell immunoglobulin and mucin-domain containing-3 (TIM-3), T-box expressed in T cells (T-bet), Thymocyte selection associated high mobility group box (Tox), Triacylglycerol (TAG), Tumour necrosis factor receptor superfamily (TNFRSF), Vitamin D receptor (VDR), Zeta-chain-associated protein kinase-70 (Zap70).

**Figure 3 vaccines-09-01392-f003:**
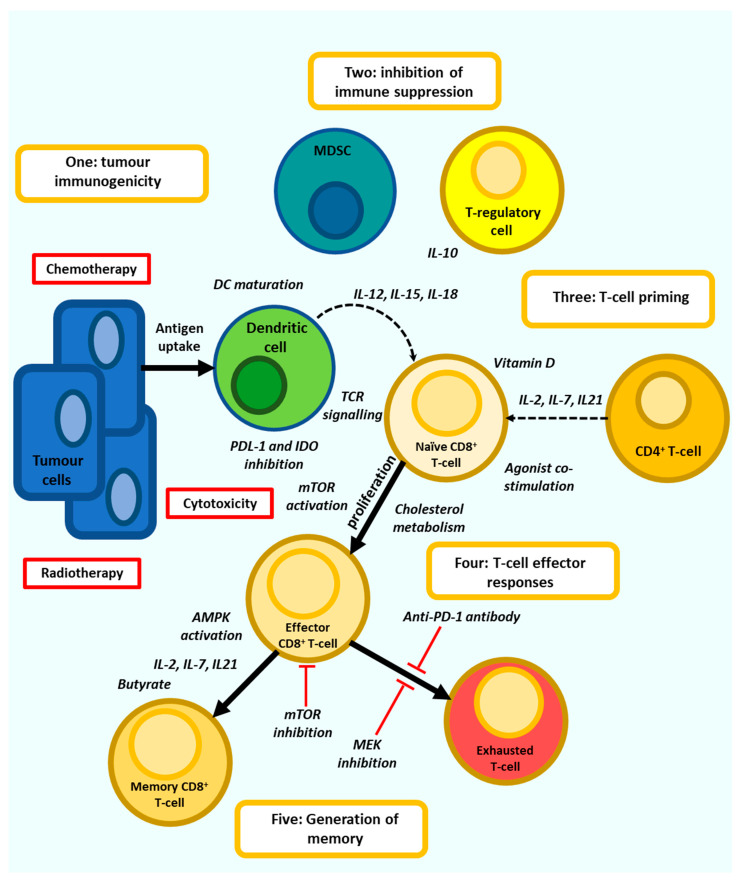
Effective cytotoxic T-cell based immunotherapy requires intervention at different stages: One: nonimmunogenic ‘cold’ tumours, typically expressing little neoantigen or downregulated HLA, will be poorly susceptible to T-cell killing. Immunogenicity of tumour can be improved through use of therapies supporting epitope spread such as radiotherapy or ICD inducing chemotherapy. Two: tumour infiltrating T-cells are subject to induction of tolerance or inhibition through suppressive features of TME or inhibitory immune subsets expressing checkpoints or cytokines such as TGF-β or IL-10 or inhibitory IDO. Three: tumour-specific CD8^+^ T-cells need to be adequately primed and capable of increased mitochondrial biogenesis to develop into efficacious effector T-cells. Four: in presence of chronic antigenic stimulation effector T-cells can develop an exhausted and dysfunctional phenotype that requires reactivation via ICI. Five: effector T-cells need to be further primed to switch from glycolysis to OXPHOS to generate long lasting anti-tumour CTL.

**Table 1 vaccines-09-01392-t001:** Categories of T-cell dysfunction.

Setting	Effect	Mechanism
Cancer	Exhaustion	Chronic antigenic exposure, altered differentiation, expression of checkpoints, and loss of effector function.
	Tolerance induction	Inhibition within the TME including T-regulatory cells MDSC and hypoxia. Tumour antigen with low affinity for cognate TCR resulting in weak T-cell activation.
Chronic inflammation	Cytokine induced dysfunction	Loss of TCR responsiveness and effector functions, bystander activation, apoptosis
Senescence	Unresponsive T-cells	Reduced production of naïve T-cells, lower expression of costimulatory receptors. Uncoupling of TCR signalling pathways and skewed T-cell differentiation.

**Table 3 vaccines-09-01392-t003:** Components of gut microbiome and dietary approaches with T-cell immune modulatory properties under investigation as cancer immunotherapies.

Microbiota	Effect on T-Cells	Questions	Refs
FMT	Can recapitulate ICB responsiveness in non-responder patients.	Nonstandardised and possibility of toxicity	[126,127]
VE800	An consortium of 11 bacteria which enhances IFN-γ expression and CD8^+^ T-cell activation in mice.	Is the constorium diverse enough for human use? A trial of VE800 and nivolumab is underway (NCT04208958)	[132]
EDP1503	Inducing systemic anti-tumour immunity by activating both innate and adaptive immunity characterised by increased cytokine expression including IFN-γ and CXCL10, activation of CD8^+^ T-cells.	What is the best combination and schedule in combination with other immunotherapies? Overall response rate of 14% across 29 patients—is a single bacterial species sufficient NCT03775850?	[133,134]
Enterococcus gallinarum	Associated with ICB responsiveness, Induces anti-tumour responses associated with an increase in the CD8^+^ T-cell:Treg ratio.	A clinical trial of MRx0518 and Pembrolizumab to treat patients with advanced solid tumours having progressed on anti-PD-1/PDL-1 monotherapy is ongoing (NCT03637803)	[120,136,137]
MET4	A greater number of MET4-associated taxa were detectable in MET4 recipients than controls.	Effect on anti-tumour immune responses not yet determined	[135]
Clostridium butyricum MIYAIRI 588	Associated with ICB responsiveness, Monotherapy with MRX0518 was able to reduce tumour size in syngeneic mouse models of breast, renal and lung carcinoma associated with an increase in the CD8^+^ T-cell:Treg ratio.	Mechanism of action involves supporting the colonisation of microbes associated with ICB response—is this the best approach? A phase I trial in combination with nivolumab plus ipilimumab in patients with metastatic RCC is underway (NCT03829111).	[138]
Ketogenic diet	Induces the induction of 3-hydroxybutyrate mediated reductions in PDL-1 expression on DC	Efficacy in a patient setting yet to be ascertained; issues with compliance?	[145]
Methionine restriction	Enhances the effects of chemotherapy in methionine dependent tumours, anti-tumour effect and potential role in T-cell activation	May also inhibit T-cell activation, potentially corrected by homocysteine supplementation however increased homocysteine is implicated in cardiovascular disease	[147,152,156]
Butyrate	Modulates T-cell activation via HDAC inhibition, associated with anti-PD-1 ICB responsiveness	Negatively correlated with the efficacy of anti-CTLA-4 ICB responsiveness	[130,140]
Inosine	Facilitates T-cell activation via aHR	Not studied in a patient setting	[149]
Dietary fibre	Increases the efficacy of anti-PD-1 therapy by altering the gut microbiome and increasing the production of SCFA including butyrate and propionate	Currently studied in murine models of colon cancer and in NSCLC patients administered high-fibre diets. The mechanism(s) of action not yet understood.	[142,143,144]

**Table 5 vaccines-09-01392-t005:** Metabolic targets under investigation as cancer immunotherapeutics.

T-Cell Metabolism	Effect on T-Cells	Questions	References
Metformin	Activates AMPK, effecting mitochondrial biogenesis and mTOR inhibition. Inhibits tumour proliferation.	May inhibit effector T-cell function by downregulating glycolysis. Sequence of Metformin administration is important (NCT03800602; NCT03709147).	[247,248,249]
Glutamine antagonism	Glutamine blockade using JHU083 inhibits tumour cell viability but results in T-cells oxidative metabolism and adopting a long-lived, highly activated phenotype	Glutamine antagonism is untested in clinical trials	[254]
L-arginine	improves anti-tumour T-cell responses by enhancing memory formation and mitochondrial respiration.	Mechanism of action not well understood	[252,253]
Butyrate	Butyrate signalling via GPCR41, preferential fuelling of oxidative phosphorylation through sustained glutamine utilization and fatty acid catabolism and ensuring that effector CD8^+^ T-cells contract in a manner that supports a pool of circulating memory cells with the necessary metabolic adaptations for long-term survival	Butyrate is positively and negatively associated with the efficacy of different ICB immunotherapies. It is unknown whether butyrate supplementation can improve cancer immunotherapy.	[130]
NAD+	Improved mitochondrial fitness upon supplementation with Nicotinamide riboside. Increased presence of TILs and survival in combination with anti-PD-1 ICB in a murine model	Direct effects on tumours are unknown. It is unknown whether supplementation is a useful strategy to improve immunotherapy	[208,250,251]
Inosine	Alternative substrate for T-cell growth and function	Some tumour cells are capable of utilising inosine as a carbon source, which may promote tumour growth	[257]

**Table 6 vaccines-09-01392-t006:** Selected list of ongoing trials combining multiple immunotherapeutic strategies.

Name	Trial/Cancer	Immunotherapies	ClinicalTrials.Gov Identifier
Interleukin-15 and -21 Armored Glypican-3-specific Chimeric Antigen Receptor Expressing Autologous T Cells as an Immunotherapy for Children with Solid Tumors (CARE)	Single group, open label, interventional study in 24 participants with paediatric solid tumours	This study will test T-cells genetically engineered with a GPC3-targeting CAR, IL15 gene and IL21 gene.	NCT04715191
Metformin Plus/Minus Fasting Mimicking Diet to Target the Metabolic Vulnerabilities of LKB1-inactive Lung Adenocarcinoma (FAME)	Interventional, nonrandomized, open-labeled, triple arm, non-comparative phase II trial 64 participants, LKB1-inactive lung adenocarcinoma	A combination of Metformin, anti-PD-1 ICB with Pembrolizumab, platinum-based immunotherapy and fasting-mimicking diet in immune suppressive, metabolically vulnerable LKB1-inactive lung adenocarcinoma.	NCT03709147
Study of GEN-1 With NACT for Treatment of Ovarian Cancer (OVATION 2)	Parallel assignment, open label, interventional study in 130 participants with ovarian cancer	GEN-1 is an IL-12 expressing plasmid being administered along with neoadjuvant chemotherapy using Carboplatin and Paclitaxel +/− subsequent GEN-1.	NCT03393884
The Effect of Diet and Exercise on ImmuNotherapy and the Microbiome (EDEN)	Parallel assignment, open label, interventional study in 80 participants with melanoma	Combination of anti-PD-1 ICB, a high fibre diet, and weekly exercise. The study will measure adherence, changes in the gut microbiome and effects on PFS and OS.	NCT04866810
Docetaxel Chemotherapy and Pembrolizumab Plus Interleukin-12 Gene Therapy and L-NMMA in Triple Negative Breast (INTEGRAL)	Single group, open label, interventional study in 30 participants with triple negative breast cancer	Combination of Interleukin 12 (IL-12) gene therapy, Methylarginine and antiPD-1 immunotherapy alongside neoadjuvant chemotherapy with docitaxel.	NCT04095689
Ketogenic Diet for Patients Receiving First Line Treatment for Metastatic Renal Cell Carcinoma (CETOREIN)	Single group, open label, interventional study in 20 participants with metastatic Renal Cell Carcinoma	Ketogenic diet and vitamin supplementation in patients treated for a metastatic renal cell carcinoma with standard of care treatment including ICB and tyrosine kinase inhibitors.	NCT04316520
All-Trans Retinoic Acid and Atezolizumab for the Treatment of Recurrent or Metastatic Non-Small Cell Lung Cancer	Single group, open label, interventional phase Ib trial in 18 participants with recurrent or metastatic NSCLC	A dose De-Escalation Study of ATRA and atezolizumab	NCT04919369
Dendritic Cell Immunotherapy Plus Standard Treatment of Advanced Renal Cell Carcinoma	Phase 2b open label, Parallel assignment interventional trial in 120 participants with advanced renal cell carcinoma	CMN-001 dendritic cell vaccine plus ICB with VEGFR kinase inhibition with Lenvatinib and mTOR inhibition with Everolimus	NCT04203901
Sirolimus and Durvalumab for the Treatment of Stage I-IIIA Non-small Cell Lung Cancer	Single group, open label, interventional study in 31 participants with NSCLC.	Inhibition of mTOR with Sirolimus for 22 days followed by anti-PDL-1 ICB using Durvalumab	NCT04348292

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
