# Peer review of "Directing T-Cell Immune Responses for Cancer Vaccination and Immunotherapy"

_vaccines, 2021, doi:10.3390/vaccines9121392_

Round 1

Reviewer 1 Report

The present review article summarizes the mechanisms for T cell dysfunction in the setting of cancer immunotherapies and describes the approaches which are currently underway to improve T cell function and the efficacy of cancer vaccination and immunotherapy. The review article is very well written, well-organised and provides all promising directions that scientists currently employ to improve T-cell efficacy for cancer immunotherapy.

Some minor changes for improvement:

  1. I would propose (if the authors agree) to put the term "T cell dysfunction" on the title, since a major part of the review article highlights in a very good and comprehensive way the numerous ways and mechanisms of T cell dysfunction.
  2. In Table 3 probiotics could be added or referred that are already under evaluation in clinical settings, often combined with checkpoint immunotherapies.
  3. in the middle of page 14 please write 13 in superscript.  (1013)
  4. please correct tumourogenesis in tumourigenesis
  5. in the first page, left column: a name is missing in Citations.

Author Response

Thank you for your review comments on my manuscript, I appreciate the time taken (considering its size) and your positive view of the work. In response to these comments I can confirm that I have added to the 'probiotics' section of the review and corrected the minor errors. Regarding the title, I gave some thought on how to include 'T-cell dysfunction' in the title but struggled to make it fit.  I think that, to some extent, the issue of dysfunction is implied in the title, and in cancer immunology more generally. I have also mentioned it in the abstract and so have decided to keep the title the same. 

Reviewer 2 Report

The manuscript by Smith et al titled “Directing T-cell immune responses for cancer vaccination and immunotherapy” discusses the mechanisms by which T-cell responses are hindered in the setting of cancer and refractive to immunotherapy, and details many of the approaches under investigation to modulate T-cell function and improve the efficacy of cancer vaccination and immunotherapy.

Overall the article is well written and interesting. The authors have thoroughly looked up the literature and cited the most recent literature. The authors covered all possible updates related to the topic. I really appreciate the efforts made by the authors and hope this article will be a good addition in the literature associated with cancer vaccination and immunotherapy.

However, the authors should include the information discussed below and also need to pay attention towards many typo errors throughout the manuscript. I would recommend the manuscript to be accepted after the following minor revision.

  1. In T cell agonists section, authors can also discuss CD40. Although it is not a direct T cell agonist but its increased antigen presentation and immunoglobulin class-switching enhances T-cell activation indirectly.
  2. Authors can also discuss the role of skin microbiome in modulating T cell immune responses.
  3. Authors should also include information about all-trans retinoic acid, a form of vitamin A, which is involved in the development and differentiation of immune-related cells and is also FDA approved for treatment of acute promyeloctic leukemia.
  4. There are typo errors throughout the manuscript that need to be corrected. A few examples are:

a) There is a typo error in fig1 legend in line 5.

b) In the gut microbiome section, in line 1 number of bacteria should be corrected. Moreover, in paragraph 4 of this section, instead of ‘A Mechanisms’ it should be ‘The mechanisms’.

c) In the T cell cytokines section, paragraph 5 have some typo errors that need to be corrected.

Author Response

Thank you for your review comments on my manuscript, I appreciate the time taken (considering its size) and your positive view of the work. In response to your comments I can confirm that:

1: I have added information related to CD40 agonists in cancer immunotherapy.

2: I have not added information related to the skin microbiome. Whilst this is interesting and certainly relevant to some cancer types such as Melanoma I wanted to keep the topic of the review restricted to general T-cell immunotherapy, rather than for specific cancers. The Gut microbiome clearly impacts many cancers via altering systemic immunity whilst the skin microbiome does not appear to have similar general immune potentiating properties.

3: I agree about the inclusion of ATRA. I initially included this in the review only to remove it in the interests of being concise. I have now included information on ATRA, placed after the section on Vitamin D and think it fits well. 

4: Thank you for detailing numerous typos. I have made every effort to identify and correct them.

Reviewer 3 Report

This review article is extensively covering almost all data available up today and literature references to formulate a specific mechanism by which T-cells responses are hindered in the setting of cancer and improve the efficacy of cancer vaccination and immunotherapy.

The specific format of this important review is exclusively designed and specifically classified by the category of responses, for the cancer therapy. This will constitute the important goals and novelty of this review paper.! The article is concluded with a rich collection of 238 references. Additionally, all 5 Tables are very informative and with concise important data with literature references. All 3 Figures are beautifully designed with excellent logistical sequences (Figure 3 with 5 stages!) for which authors should be congratulated.

            The following suggested changes and recommendations should be introduced before the publication of the manuscript.

  1. Page 1. Keywords. Please add two more keywords to the limit of six.
  2. Page 28 line 8, the last sentence “These agents are …” is too long and must be split in order to keep the flow of information in logistical and grammatical order.
  3. Should the short list of abbreviations at the end of this review be considered? It is up to the Editor to decide. On the positive note, such addition will help to follow the flow of information in more comprehensive way.

The manuscript is of very good quality and urgent importance and is very well written and edited in order to meet the standard for the articles published in Vaccines. Thus, I certainly recommend it for publication after the correction of these suggested minor changes and recommendations.

Author Response

Thank you for your review comments on my manuscript, I appreciate the time taken (especially considering its size) and your very positive view of the work. My review took significant time and effort to write and it was gratifying to read that it has been positively received. In response to your comments I can confirm that:

1: I have added additional keywords.

2: I agree that the last sentence needed to be rewritten and think I have improved upon this part.

3: I have asked the editor whether they think a list of abbreviations is suitable. I can see the advantages of this but have made an effort to include them in the text and figures.